# Assembly of transmembrane pores from mirror-image peptides

Smrithi Krishnan R ®[1,2,6], Kalyanashis Jana[3,6], Amina H. Shaji[1,6], Karthika S. Nair[4,5], Anjali Devi Das ®[1], Devika Vikraman[1,2], Harsha Bajaj ®[4,5], Ulrich Kleinekathöfer ®[3] & Kozhinjampara R. Mahendran ®[1]✉

Tailored transmembrane alpha-helical pores with desired structural and functional versatility have promising applications in nanobiotechnology. Herein, we present a transmembrane pore DpPorA, based on the natural pore PorACj, built from D-amino acid α-helical peptides. Using single-channel current recordings, we show that DpPorA peptides self-assemble into uniform cation-selective pores in lipid membranes and exhibit properties distinct from their L-amino acid counterparts. DpPorA shows resistance to protease and acts as a functional nanopore sensor to detect cyclic sugars, polypeptides, and polymers. Fluorescence imaging reveals that DpPorA forms well-defined pores in giant unilamellar vesicles facilitating the transport of hydrophilic molecules. A second D-amino acid peptide based on the polysaccharide transporter Wza forms transient pores confirming sequence specificity in stable, functional pore formation. Finally, molecular dynamics simulations reveal the specific alpha-helical packing and surface charge conformation of the D-pores consistent with experimental observations. Our findings will aid the design of sophisticated pores for single-molecule sensing related technologies.

Membrane-spanning protein pores have been widely engineered for the single-molecule sensing of biomacromolecules, including nucleic acid sequencing, which will open opportunities in personalized diagnosis[1–5]. With more recent developments in protein engineering, a wide range of natural pores and synthetic DNA pores have been reported for sensing applications[6–17]. However, engineered alpha-helical pores for single-molecule sensing remain less explored[18–24]. This is primarily because the engineering of alpha-helical pores often distorts their functional structure as multiple side-chain interactions within individual helices are required to stabilize their folding and assembly[25–28]. Nevertheless, there is tremendous interest in building α-helical transmembrane pores since they offer a wide range of functional properties similar to natural membrane proteins[29–32]. Engineering beta-barrels has produced only slight changes in ion selectivity, whereas engineered alpha-helical pores

have the potential to mimic highly selective natural ion channels[1,29–32]. Importantly, discovering alpha-helical pores with defined structures that selectively conduct ions and molecules could lead to the development of new functional nanopore biosensors[1,33].

To tackle this challenge, the de novo design of alpha-helical barrels has been employed to identify appropriate sequences and construct barrels of defined structure and geometry[26,34,35]. Coiled-coil structural domains have been targeted to construct water-soluble channels of defined structural conformations, diverse oligomerization states, and stable topologies[19,20,36]. Most recent efforts focused on converting soluble forms of de novo pores to transmembrane structures by targeting specific amino acids aided by computational design[28,31,32]. Recently, advanced computational designs have been introduced to build tailored transmembrane pores for various

[1]Membrane Biology Laboratory, Transdisciplinary Research Program, Rajiv Gandhi Centre for Biotechnology, Thiruvananthapuram 695014, India. [2]Manipal Academy of Higher Education, Manipal, Karnataka 576104, India. [3]Department of Physics and Earth Sciences, Jacobs University Bremen, 28759 Bremen, Germany. [4]Microbial Processes and Technology Division, CSIR- National Institute for Interdisciplinary Science and Technology (NIIST), Thiruvananthapuram 695019, India. [5]Academy of Scientific and Innovative Research (AcSIR), CSIR-Human Resource Development Centre, Ghaziabad 201002, India. [6]These authors contributed equally: Smrithi Krishnan R, Kalyanashis Jana, Amina H Shaji. ✉e-mail: mahendran@rgcb.res.in

applications[31,32]. However, none of the designed channels formed pores of high ion conductance and charge selectivity in single-channel electrical recordings[22,31,32,37]. An alternative approach is to build and engineer transmembrane alpha-helical pores based on natural alpha-helical assemblies such as Wza and PorACj[22,30,38,39]. Such pores can be built using simple chemical synthesis and have considerable advantages over natural membrane pores. Of these examples, the synthetic cWza peptide pore exhibited low conductance and intermediate states[22]. In contrast, pPorA peptides self-assembled into uniform oligomers that spontaneously inserted into lipid bilayers to form functional alpha-helical pores[29,30]. Notably, this is one of the rare channels to show very large conductance of tunable selectivity exploited for sensing charged peptides[30]. The incorporation of unnatural amino acids into such peptides inspired by natural systems during synthesis may result in sophisticated pores that exhibit structural and functional versatility[40]. Incorporating D-amino acids may also result in increased biostability due to resistance to protease enzymes and therefore expand the scope of developing new therapeutics and drug delivery systems[1,41].

In this study, we constructed stereo-inverted alpha-helical pores based on the pores PorACj and Wza. Single-channel recordings revealed that both pores exhibited distinct structural and functional properties compared to their L counterparts. In addition to the free-standing membranes used in the current recordings, the pore formation and functional assembly in giant vesicles was confirmed using fluorescence imaging. Molecular dynamics simulations of the modeled pores revealed a stable helical conformation and ion conduction pathway. We suggest that this new class of mirror-image peptide pores with large conductance and selectivity will likely find applications in nanopore technology.

## Results

### Biophysical and electrical properties of DpPorA

DpPorA peptides containing D-amino acids based on the natural membrane pore PorACj were synthesized using solid-phase peptide synthesis and purified by reversed-phase high-performance liquid chromatography (HPLC) (Fig. 1a, Supplementary Fig. 1 and Supplementary text). Furthermore, their mass was evaluated by mass spectrometry (Supplementary Fig. 1). The circular dichroism spectra of DpPorA peptides in n-dodecyl β-D-maltoside (DDM) micelles confirmed the alpha-helical conformation of the peptides with equivalent but opposite ellipticity to that of LpPorA peptides (Fig. 1a). The CD spectra suggest that the peptides were indeed chiral isomers. DpPorA peptides in SDS polyacrylamide gel electrophoresis (PAGE) showed a ~35 kDa band corresponding to stable octameric oligomers, indicating preoligomerization (Fig. 1b). Since the octameric preoligomers are stable in SDS-PAGE, the corresponding band (~35 kDa) was extracted from the gel to examine their single-channel functional properties (Supplementary text). The gel extracted DpPorA preoligomers inserted into 1,2-diphytanoyl-sn-glycero-3-phosphocholine (DPhPC) planar lipid bilayers at +50 mV and +100 mV to form stable pores (Fig. 1c, d and Supplementary Fig. 2). We observed multiple single-channel insertion events ($n = 75$) at +100 mV and obtained unitary conductance histograms, which revealed the homogeneity of the pores. DpPorA exhibited a mean unitary conductance (G) of $1.5 \pm 0.3$ nS at +100 mV in 1 M KCl (Fig. 1e). Notably, DpPorA showed gating and sub-conductance states with noisy ion current openings and multiple downward spikes at higher voltages (>±100 mV), whereas it remained in open conductance states at +50 mV (Fig. 1c, d and Supplementary Fig. 2). The pore conductance varied linearly with applied voltage and selectivity measurements indicated that the DpPorA peptides formed cation-selective pores with a permeability ratio of $P_{K^+}/P_{Cl^-} = \sim7:1$ (Fig. 1f, Supplementary Fig. 3 and Supplementary text). The statistical analysis of 100 independent single DpPorA insertion events revealed that 75% of the DpPorA population exhibited conductance of ~1.5 nS in 1 M KCl

referred to as (S) while a minor population (25%) possessed a conductance of ~4 nS referred to as (L) (Fig. 1g–i and Supplementary Fig. 3). The single-channel conductance of DpPorA (S) increased in high salt electrolyte buffer (3 M KCl) and the pore remained in a stable conductance state under different salt conditions (Supplementary Fig. 4). LpPorA formed stable pores of high unitary conductance ~4 nS in 1 M KCl and exhibited single-channel properties distinct from DpPorA (Supplementary Fig. 4).

### DpPorA as a single-molecule nanopore sensor

We explored the possibility of using DpPorA as a functional single-molecule nanopore sensor to characterize various biomolecules. The pore was used to detect various positively charged analytes due to its cation-selective nature (Fig. 1f). At first, we examined the interaction of the 8-fold symmetric cationic cyclodextrin, am$_8$γCD, with DpPorA (S) at single-molecule resolution (Fig. 2a and Supplementary Fig. 5). The addition of 10 μM am$_8$γCD to the cis side of the pore resulted in complete closure (100% block) at negative voltages, resulting in a steady closed conductance state (Fig. 2a). The negative applied voltage serves as the driving force to electrophoretically pull cationic CDs into the pore promoting electrostatic interactions with negatively charged residues in the pore lumen (Supplementary Fig. 5). The voltage-driven electrostatic binding of the CDs to DpPorA is affirmed by the increase in the rate of pore closure with increasing applied voltage (Supplementary Fig. 5). Consistent with this, no blockages were observed at the positive voltages as the cationic CDs are electrostatically repelled (Supplementary Fig. 5). Next, we perfused the cyclodextrins added to the cis side and added 10 μM am$_8$γCD to the trans side of the same pore (Fig. 2a and Supplementary Fig. 5). At positive voltages, am$_8$γCD produced complete pore closure and no blockages were observed at negative voltages, indicating electrostatic binding of cationic CDs with the pore surface (Fig. 2a and Supplementary Fig. 5). The statistical analysis of CD binding with DpPorA revealed that am$_8$γCD blocks the cis side of the pore more rapidly than the trans side (Fig. 2a, Supplementary Fig. 5 and 6). The pore remained in the open state for a shorter duration on cis side am$_8$γCD addition, suggesting a clear-cut asymmetry in cationic CD binding (Fig. 2a, Supplementary Figs. 5 and 6). Based on this data, we suggest that the cis side of the pore consists of abundant negatively charged residues leading to strong electrostatic CD binding. We observed a similar CD binding pattern in 95% of the cases ($n = 24$ of 25 experiments) and thus, the possible unidirectional orientation of the pore in the lipid membrane was identified (Supplementary Fig. 6). Furthermore, we examined the interaction of the pore with both neutral γCD and anionic s$_8$γCD, which did not produce any current blockages, confirming the dominant negatively charged pore surface (Fig. 2b and Supplementary Fig. 6). Remarkably, am$_8$γCD binds to DpPorA with a higher affinity than LpPorA. For example, at +50 mV, am$_8$γCD completely blocked DpPorA ($n = 3$) whereas it produced time-resolved ion current blockages through LpPorA with a mean blockage time ($\tau_{off}$) of $0.22 \pm 0.03$ ms ($n = 3$) (Fig. 2a and Supplementary Fig. 4). We suggest that the different binding kinetics of am$_8$γCD to DpPorA and LpPorA occurs because the affinity site of the D pore is altered as the stereo inversion of the amino acids results in distinct pore conformation.

Furthermore, we investigated the binding of cationic polypeptides through DpPorA and quantified the translocation kinetics (Fig. 2c, Supplementary Fig. 7 and Supplementary Table 1). The cationic polypeptide nonaarginine, R9 (10 μM, cis), interacted with DpPorA and produced time-resolved ion current blockages specifically at negative voltages (Fig. 2c). Notably, the mean residence time of R9 blocking ($\tau_{off}$) was calculated at different voltages, where the increasing dissociation rate ($k_{off} = 1/\tau_{off}$) with increasing negative voltages indicated R9 translocation (Fig. 2c, Supplementary Fig. 7 and Supplementary Table 1). Similarly, the time between successive R9 blocking ($\tau_{on}$) was calculated at different voltages. Here, the association rate

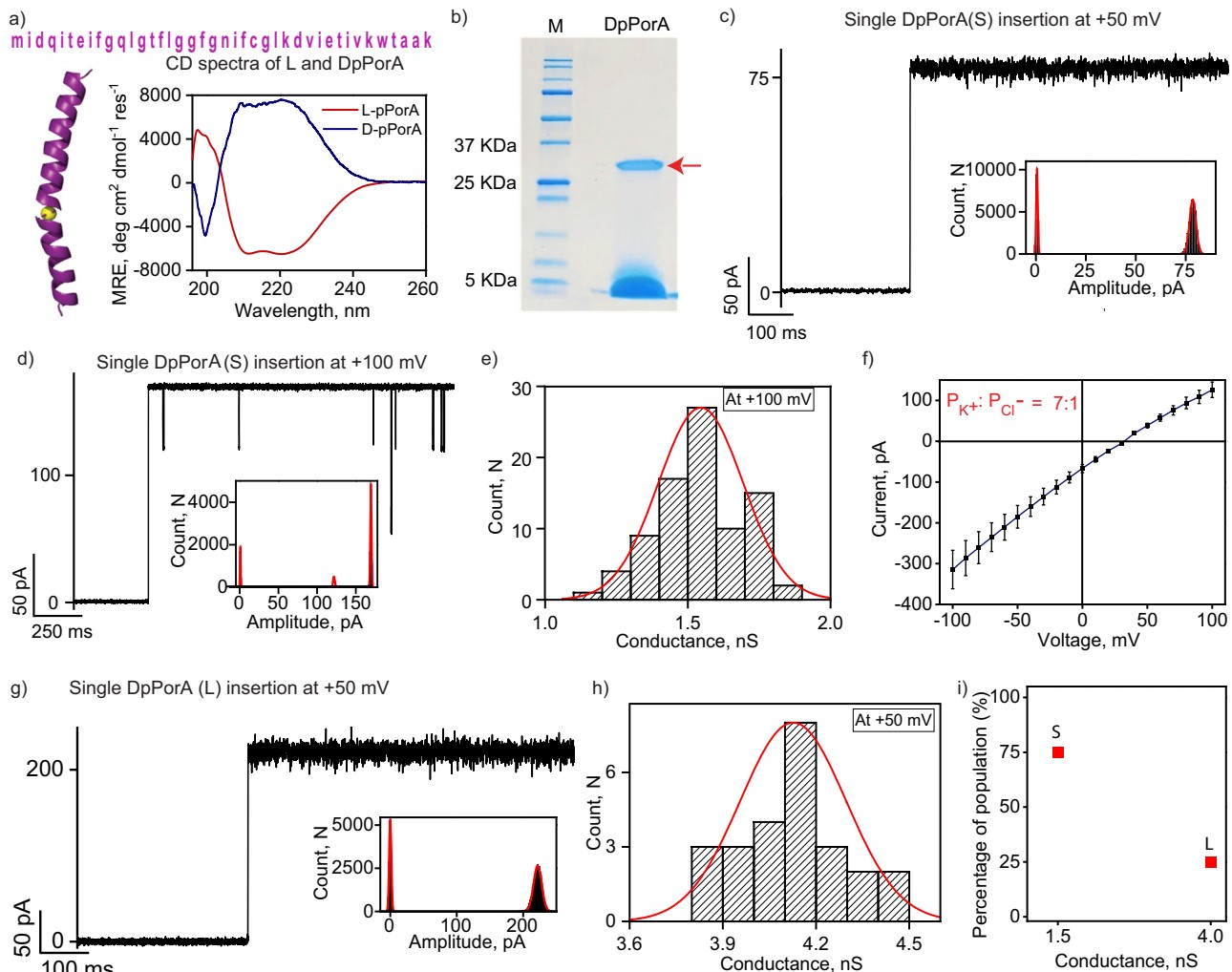

**Fig. 1 | Functional assembly and electrical properties of DpPorA. a** DpPorA peptide sequence and CD spectra for 100 μM DpPorA (blue) and LpPorA (red) in phosphate-buffered saline with 1% DDM. **b** DpPorA peptides run on SDS-PAGE showing self-assembled preoligomers (red arrow) and monomers. Data are representative of more than five repeats. **c** Single DpPorA insertion (S) at +50 mV and corresponding current amplitude histogram. **d** Single DpPorA insertion (S) at +100 mV and corresponding current amplitude histogram. **e** The mean unitary conductance histogram for DpPorA insertion events (S) at +100 mV was obtained by fitting the distribution to a Gaussian (number of insertion events, *n* = 75). **f** I–V curve of a single DpPorA to determine reverse potential and charge selectivity. Error bars represent 15% standard error mean between 4 independent experiments. **g** Single DpPorA insertion at +50 mV showing a large conductance state (L) and corresponding current amplitude histogram. **h** The mean unitary conductance histogram for DpPorA insertion events (L) at +50 mV was obtained by fitting the distribution to a Gaussian (number of insertion events, *n* = 25). **i** Plot representing two distinct DpPorA species (S and L). Electrolyte: 1 M KCl, 10 mM HEPES, pH 7.4 except **f** 1 M KCl, cis and 0.15 M KCl, trans. The current signals were digitally filtered at 2 kHz. Source data are provided as a Source Data file.

($k_{on}$ = $1/\tau_{on}$·C) decreased with increasing negative voltage, indicating faster translocation of R9 at higher voltages (Fig. 2c, Supplementary Fig. 7 and Supplementary Table 1). This data suggest that the applied voltage drives the translocation of R9 peptides, despite their large size, through DpPorA (Fig. 2c and Supplementary Fig. 7). A similar binding pattern of R9 with DpPorA was observed on the trans side addition at positive voltages (Supplementary Fig. 7). The addition of the smaller peptide tetraarginine, R4 (10 μM, cis), to the pore produced short, less frequent blockage events at negative voltages exclusively in low salt electrolyte buffer consistent with the large size of the pore (Supplementary Fig. 7). As noted above, DpPorA is cation-selective and anionic nonaaspartate (D9) did not produce any ion current blockages, indicating negligible binding of D9 to the pore (Fig. 2c and Supplementary Fig. 7).

We then explored the use of DpPorA pores as nanoreactors for single-molecule covalent chemistry. For example, the cysteines at the 24th position in the ion conduction pathway can act as sites for site-specific chemical modification using activated PEG thiol blockers

(Fig. 2d and Supplementary Fig. 8). We initially recorded the conductance of the DpPorA in the absence of PEG thiol blockers at +50 mV and +25 mV as the pore remained in a fully open state (Supplementary Fig. 8). The addition of 1 mM monomethoxy poly (ethylene glycol)-*o*-pyridyl disulfide) (MePEG-OPSS-1k) to the cis side of the DpPorA produced a stepwise pore closure at +50 mV and +25 mV, indicating covalent modification of the cysteines (disulfide bond, *n* = 4) similar to that observed for LpPorA[29] (Fig. 2d and Supplementary Fig. 8). The addition of 10 mM dithiothreitol resulted in cleavage of the disulfide bonds and the pore reverted to its open conductance state, demonstrating the high specificity of the covalent modification (Fig. 2d and Supplementary Fig. 8). We have also added am$_8$γCD (10 μM, trans) to DpPorA prior to MePEG-OPSS-1k addition. Characteristic blockages confirmed pore functionality, after which the am$_8$γCD was completely perfused and MePEG-OPSS-1k was added to the same pore, leading to stepwise pore closure (Supplementary Fig. 8). Interestingly, no blockage was observed upon adding a larger thiol blocker MePEG-OPSS-5k. The molecule cannot penetrate DpPorA to access cysteine

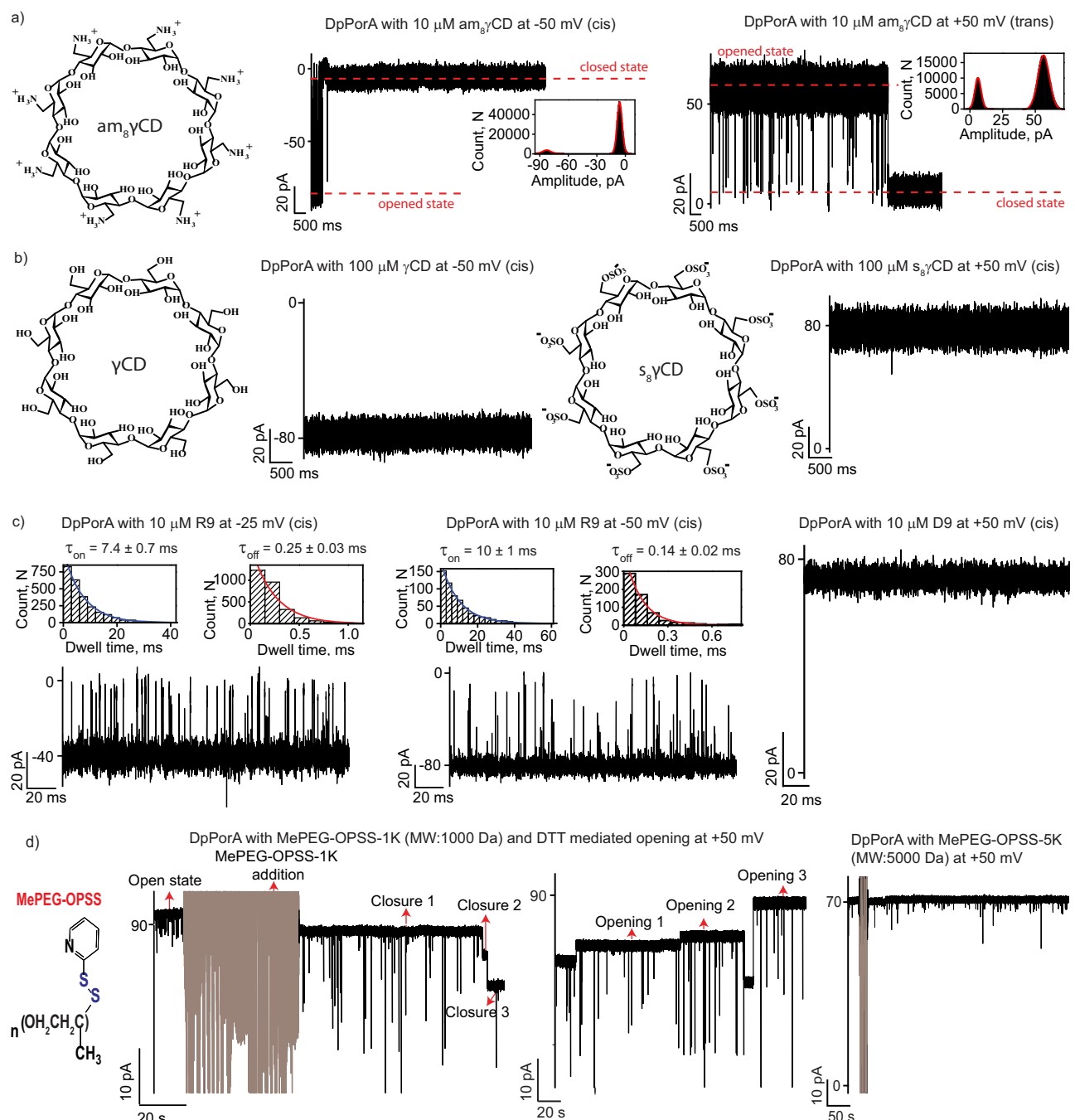

**Fig. 2 | DpPorA for single-molecule sensing. a** Chemical structure of cationic am$_8$γCD. Interaction of am$_8$γCD with single DpPorA (10 μM, cis) at −50 mV and (10 μM, trans) at +50 mV. Inset shows the corresponding current amplitude histogram. **b** Chemical structure of neutral γCD and interaction with single DpPorA (100 μM, cis) at −50 mV. Chemical structure of anionic s$_8$γCD and interaction with single DpPorA (100 μM, cis) at +50 mV. **c** Interaction of nonaarginine (R9) with single DpPorA (10 μM, cis) at −25 mV and −50 mV. Insets show corresponding dissociation dwell time (τ$_{off}$) and association dwell time (τ$_{on}$) dwell time histograms

of R9 blocking fitted with a monoexponential probability function. Interaction of nonaaspartate (D9) with single DpPorA (10 μM, cis) at +50 mV (left). **d** Electrical recordings showing the reversible chemical modification of 1 mM MePEG-OPSS-1k with DpPorA at +50 mV and the addition of 10 mM DTT resulted in the pore opening. Electrical recordings showing no chemical modification of 1 mM MePEG-OPSS-5k with DpPorA at +50 mV. The current signals (**a, b**) were filtered at 2 kHz and sampled at 10 kHz. The current signals (**c**) were digitally filtered at 7 kHz. The current signals **d** were digitally filtered at 500 Hz.

residues, owing to the significantly larger hydrodynamic radius of MePEG-OPSS-5k (Fig. 2d). This result further highlights the constricted pore size of the DpPorA compared to the L-counterpart that is blocked by MePEG-OPSS-5k[29].

## Functional stability of DpPorA pores to protease
Next, using SDS-PAGE and single-channel recordings, we investigated the stability of DpPorA in the presence of proteinase K, a protease

enzyme that displays broad substrate specificity (Fig. 3 and Supplementary Fig. 9). DpPorA peptide was first treated with proteinase K and the effect of proteolysis was determined at optimized reaction conditions (Supplementary text). Proteinase K treated and untreated DpPorA peptides were subjected to SDS-PAGE to study the protease reaction profile (Fig. 3a). Remarkably, DpPorA peptides were highly resistant to proteinase K and showed a well-defined monomeric band and a preoligomer band similar to those observed for untreated

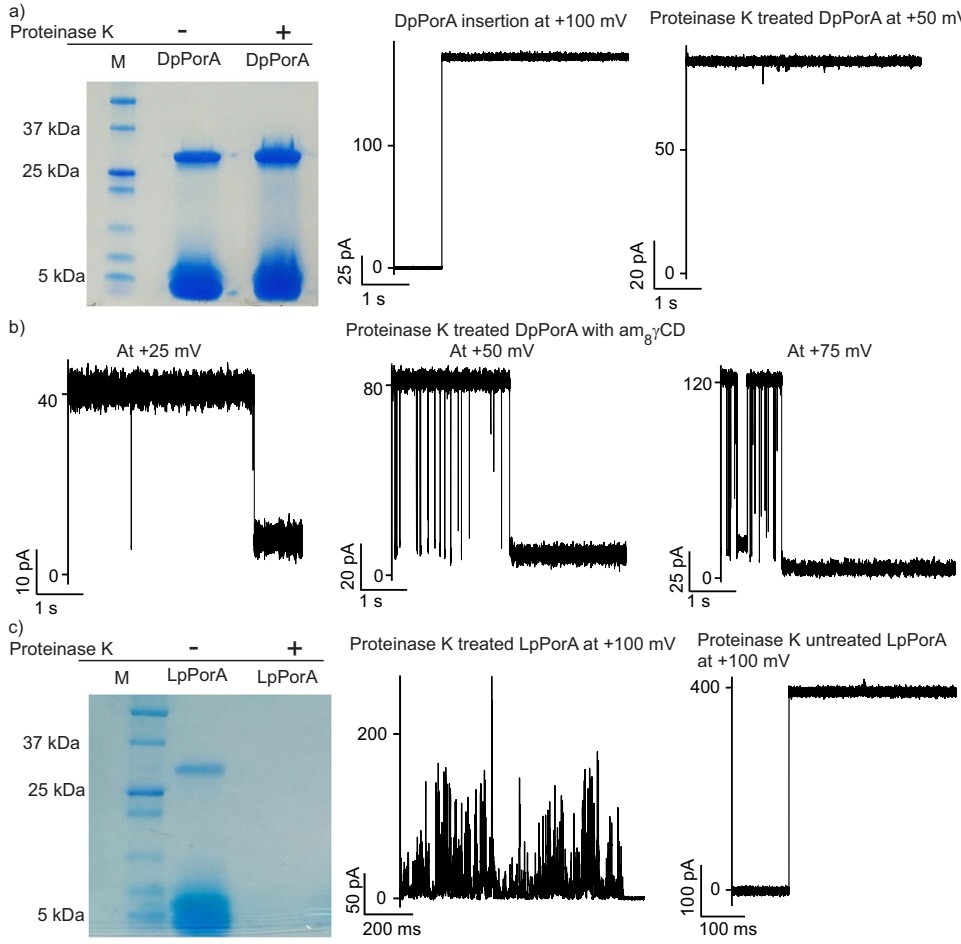

**Fig. 3 | Functional stability of DpPorA to protease reaction. a** DpPorA peptides treated with proteinase K run on SDS-PAGE showing monomer and pre-oligomer band. Data are representative of more than three repeats. Single stable pore insertion of gel extracted proteinase K treated DpPorA peptides at +100 mV and single DpPorA in stable conductance state at +50 mV. **b** Interaction of proteinase K reacted DpPorA with am$_8$γCD (10 μM, trans) at +25 mV, +50 mV and +75 mV. **c** Proteinase K treated and untreated LpPorA peptides run on SDS-PAGE. Data are representative of more than three repeats. Proteinase K treated LpPorA producing current bursts at +100 mV and single pore insertion of proteinase K untreated LpPorA at +100 mV. All current signals were filtered at 2 kHz and sampled at 10 kHz. Source data are provided as a Source Data file.

DpPorA (Fig. 3a and Supplementary Fig. 9). The proteinase K treated DpPorA preoligomers were extracted from the gel to investigate their electrical and functional properties. Interestingly, the DpPorA pre-oligomers inserted into DPhPC bilayers and formed stable pores of unitary conductance ~1.5 nS in 1 M KCl ($n = 25$) (Fig. 3a and Supplementary Fig. 9). Importantly, the proteinase K treated DpPorA pre-oligomers exhibited single-channel electrical properties identical to that of the untreated DpPorA preoligomers (Figs. 3a and 1c). Furthermore, the addition of am$_8$γCD to proteinase K treated DpPorA resulted in pore blocking in a voltage-dependent manner, indicating the presence of structurally stable functional pores (Fig. 3b). In contrast, LpPorA peptides treated with proteinase K showed no distinguishable bands in SDS PAGE and did not form pores in the lipid bilayers (Fig. 3c and Supplementary Fig. 9). Occasionally, we observed unstable, noisy pores with fluctuating conductance states showing non-specific peptide–membrane interactions (Fig. 3c and Supplementary Fig. 9). As expected, untreated LpPorA peptides displayed bands in SDS-PAGE corresponding to both the monomeric and octameric preoligomer species (Fig. 3c and Supplementary Fig. 9). These peptides readily inserted into the lipid bilayers forming stable uniform pores of ~4 nS conductance at +100 mV in 1 M KCl (Fig. 3c). The above results suggest that unlike DpPorA peptides, which form stable pores after proteinase K treatment, LpPorA peptides are readily degraded by proteinase K, inhibiting pore formation.

## Transport across DpPorA pores in giant unilamellar vesicle system

The channel-forming activity of DpPorA peptides was further examined in giant unilamellar vesicles (GUVs), an alternative membrane model system employed to investigate the pore-forming properties of membrane proteins[42–44]. We reconstituted DpPorA peptides in GUVs and investigated their channel formation using diffusion-based uptake assays utilizing fluorescence detection (Fig. 4). The DDM solubilized peptides were added to pre-formed GUVs, and the uptake of the hydrophilic fluorescent dye Alexa-Fluor 350 across the vesicles was then monitored over time (Fig. 4). The fluorescent molecule Alexa-Fluor 350 hydrazide (Molecular Weight = 349 Da) is a hydrophilic molecule that does not traverse the membrane without pore-forming membrane proteins, as shown in control vesicles (Fig. 4a). In contrast, upon the addition of DpPorA, we observed increased uptake of the dye inside single vesicles over time, as represented by increasing fluorescence intensity in the presence of peptides compared to the control (Fig. 4b). The fluorescence intensity inside ($I_{in}$) the vesicles was quantified and normalized against the intensity outside ($I_{out}$) the vesicles for a time-dependent analysis of dye transport (Supplementary text). We observed an increase in normalized fluorescence intensity inside the vesicles ($\frac{I_{in}}{I_{out}}$) over time in peptide reconstituted vesicles indicating flux of the Alexa dye across the protein inside the lumen of the vesicle (Fig. 4c). On the contrary, in control vesicles to

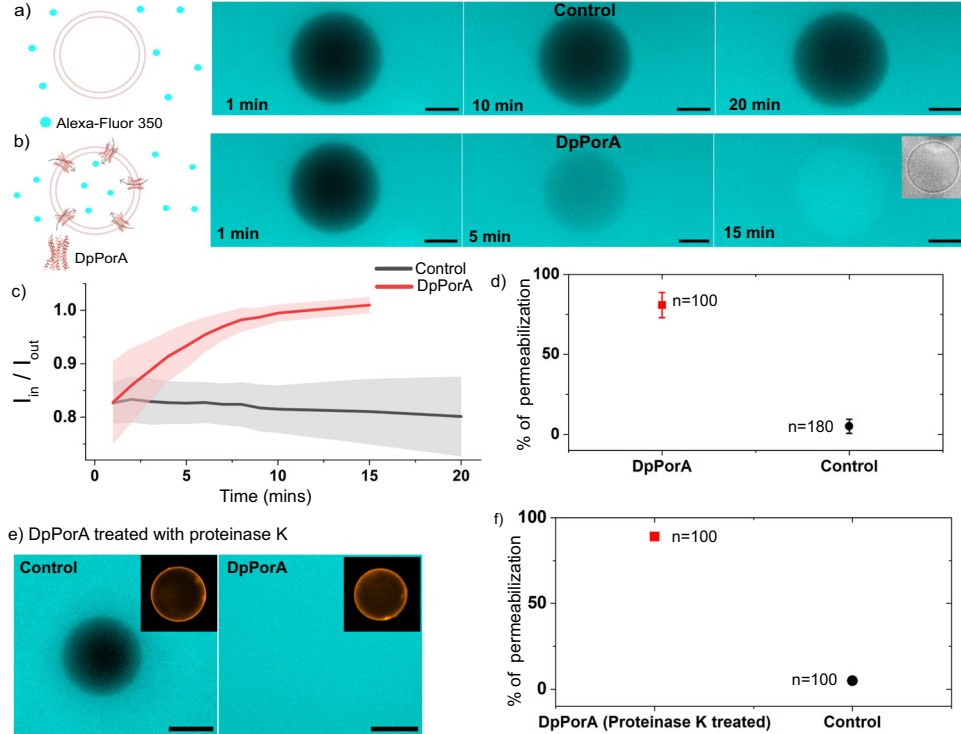

**Fig. 4 | Functional assembly of DpPorA in giant unilamellar vesicles.** Schematic and representative fluorescence image of single vesicle outlining the transport of molecules in a time-dependent manner in the **a** absence of a pore and **b** in the presence of DpPorA, where false blue color represents the Alexa-Fluor 350 dye in vesicles. Inset shows the phase-contrast image of the vesicle. **c** Time-dependant curve of normalized intensity of single vesicles with DpPorA ($n = 24$ individual vesicles) and without peptide ($n = 10$ individual vesicles) is represented over time; red and black lines represent the mean values. The shaded region (error bands) represents the corresponding standard error of the mean. **d** Vesicle permeabili-zation percentage is shown in the absence and presence of DpPorA. Permeabili-zation percentage of DpPorA is 80.9 ± 7.9% (red, mean ± SD from $n = 100$ vesicles

and $N = 5$ independent batches) and control is 3.2 ± 3% (black, mean ± SD from $n = 180$ vesicles and $N = 5$ independent batches). **e** Representative fluorescence image of single vesicle in the absence and presence of DpPorA incubated with proteinase K for 15 min displaying dye transport. The false blue color represents the Alexa-Fluor 350 dye in vesicles. Insets display the labeled vesicles in fluorescence. **f** Vesicle permeabilization percentage in the presence of DpPorA treated with proteinase K. Permeabilization percentage of DpPorA is 88 ± 1.4% (red, mean ± SD from $n = 100$ vesicles and $N = 2$ independent batches). Permeabilization percentage of control is 5 ± 1.4% (black, mean ± SD from $n = 100$ vesicles and $N = 2$ independent batches). Buffer conditions: 100 mM KCl, 10 mM HEPES pH 7; scale bar: 10 μm. The schematics are created with BioRender.com.

which only detergent was added, the normalized fluorescence inten-sity remained constant or decreased (due to bleaching) over an extended time period (Fig. 4c). We also calculated the percentage population of vesicles that displayed complete dye permeabilization in the presence and absence of peptides for statistical analysis of their channel-forming properties. Complete permeabilization of dye inside vesicles with peptides is indicated by $\frac{I_{in}}{I_{out}} > 0.99$, where the fluorescence intensity of the dye inside the vesicles is almost equal to that outside ($I_{in} \approx I_{out}$). The vesicle permeabilization percentage in the presence of DpPorA (0.5 to 1 μM) was calculated to be 80.9 ± 7.9%, whereas that of the control was only 3.2 ± 3% (Fig. 4d). The significantly lower popu-lation of permeabilized vesicles in control suggests that some vesicles are inherently leaky or leaky due to the action of detergent added to control vesicles. Based on this, we conclude that DpPorA peptides form pores that facilitate the transport of small hydrophilic molecules across the vesicle membrane. A similar trend of transport of Alexa-Fluor 350 was observed in LpPorA reconstituted in GUVs, confirming pore formation (Supplementary Fig. 10). Furthermore, we checked the activity of DpPorA after proteinase K treatment and observed the transport of Alexa dye inside the vesicle lumen represented by fluor-escence intensity $I_{in} \approx I_{out}$ (where the vesicles show fluorescence inside their lumen), as compared to control (Fig. 4e). The vesicle permeabi-lization percentage was estimated to be 88 ± 1.4%, whereas that of the control was only 5 ± 1.4% (Fig. 4f). We conclude that DpPorA can form functional pores even after treatment with proteinase K, demonstrat-ing its stable structural conformation.

## Assembly of transient DcWza pores

We also examined the pore-forming properties of a D-amino acid pep-tide corresponding to the *E. coli* polysaccharide transporter Wza (DcWza) to demonstrate specificity in the assembly of stable pores (Fig. 5). The DcWza peptides formed pores into DPhPC bilayers and exhibited multiple downward spikes at a high applied voltage of ±200 mV (Fig. 5a and Supplementary Fig. 11). The mean unitary con-ductance of the pore in the open state was calculated to be 0.95 ± 0.1 nS at +200 mV in 1 M KCl based on multiple pore insertions ($n = 50$) and existed in the open conductance state at +50 mV (Fig. 5a). Interestingly, these pores did not stay in an open conductance state and ejected from the membrane with downward ion current spikes (Fig. 5b and Supple-mentary Fig. 11). The duration of the open conductance states varied from several seconds to minutes, confirming the formation of transient DcWza pores (Fig. 5b and Supplementary Fig. 11). The stepwise pore insertion and ejection can be distinguished, demonstrating the clear assembly and disassociation of the DcWza pores in the membrane (Supplementary Fig. 11). Notably, no blockages were observed upon the addition of am₈γCD (100 μM, trans) to the DcWza pore when it remained in the open state (Supplementary Fig. 11). In addition, DcWza peptides formed fluctuating pores that predominantly remained closed and occasionally formed noisy, unstable pores of varying conductance states (Fig. 5b and Supplementary Fig. 12). Statistical analysis revealed that DcWza peptides produced three distinct pore populations in the membrane (Supplementary Fig. 12). Additionally, DcWza pore forma-tion and subsequent dye transport was investigated in giant vesicles.

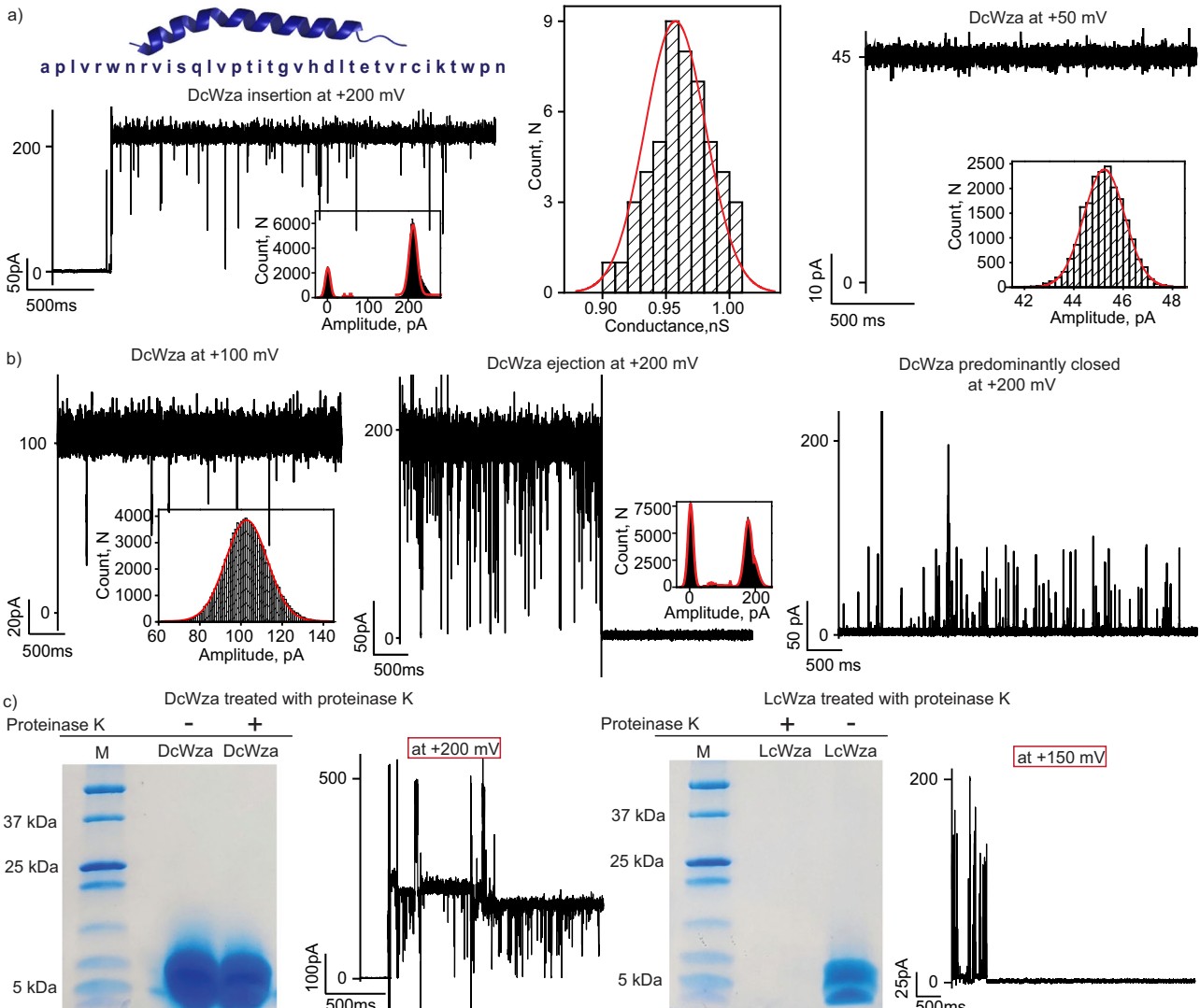

**Fig. 5 | Single-channel properties of DcWza. a** DcWza peptide sequence, single DcWza insertion at +200 mV and corresponding current amplitude histogram. The mean unitary conductance histogram for DcWza insertion at +200 mV was obtained by fitting the distribution to a Gaussian (*n* = 50). Single DcWza in a stable conductance state at +50 mV and corresponding current amplitude histogram. **b** Single DcWza at +100 mV and ejection of DcWza from the membrane at +200 mV with corresponding current amplitude histogram.

DcWza in its closed conductance state showed spike events at +200 mV **c** The DcWza peptides treated with proteinase K run on SDS PAGE, showing monomers that formed transient pores in the lipid bilayers. The LcWza peptides treated by proteinase K run on SDS-PAGE did not form stable pores in lipid bilayers. SDS gel are representative of more than three repeats. The current signals were filtered at 2 kHz and sampled at 10 kHz. Source data are provided as a Source Data file.

There was no dye transport in DcWza after 1 h of incubation with the dye and the vesicle permeabilization percentage was calculated to be 4.6 ± 2.5%, which was similar to the control permeabilization percentage of 4.2 ± 2.1% (Supplementary Fig. 12). This data indicates no stable pore formation in agreement with single-channel electrical recordings. Notably, the LcWza peptides formed stable uniform pores of unitary conductance ~0.75 nS in 1 M KCl that remained in the lipid bilayer without ejection (Supplementary Fig. 13). The contrasting single-channel properties of LcWza and DcWza demonstrate the effect of amino acid stereo inversion in stable pore formation (Supplementary Fig. 13). Upon treatment with proteinase K, DcWza peptides showed

bands corresponding to the monomer in the SDS PAGE and formed transient pores identical to those formed by untreated DcWza peptides (Fig. 5c and Supplementary Fig. 13). In contrast, LcWza peptides treated with proteinase K did not show any detectable band on SDS PAGE and no pore formation was observed, confirming peptide degradation (Fig. 5c and Supplementary Fig. 13). In summary, while the differences in pore formation of DpPorA and DcWza demonstrate the importance of sequence specificity in stable, functional pore formation, both mirror pores are stable to protease, demonstrating the advantages of stereo inversion to enhance the proteolytic resistance of pore-forming peptides.

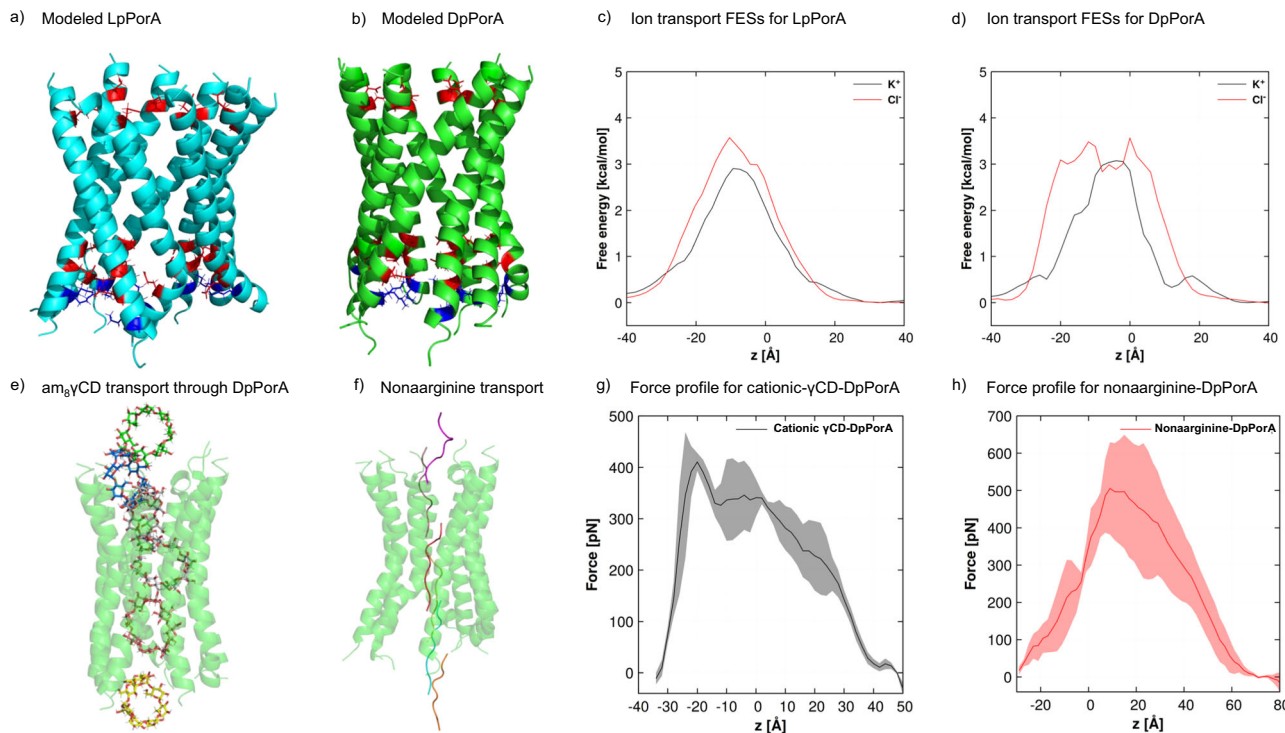

**Fig. 6 | Structures of the designed pores after 200-ns-long unbiased MD simulations and permeation of analytes through DpPorA. a, b** Stable LpPorA (cyan) and DpPorA (green) structures in cartoon representation. Basic amino acid residues are shown in blue, while acidic residues are shown in red. **c** 1D multi-ion FESs along the LpPorA channel axis *z* for K⁺ and Cl⁻ ions. The FESs has been estimated from unbiased simulations in the presence of 1 M KCl at 300 K. **d** 1D multi-ion FESs for DpPorA. **e** Transport of am₈γCD through DpPorA and **f** Nonaarginine transport through DpPorA. Average force profile from steered MD simulation for the permeation of **g** am₈γCD and **h** nonaarginine as a function of reaction coordinate z, i.e., the center of mass distance between am₈γCD or nonaarginine and the $C_\alpha$ atoms of DpPorA and the shaded error bars represent the standard deviations.

## Pore conformation and ion transport revealed by molecular dynamics simulations

Models of the investigated pores were constructed based on their respective amino acid sequences (Supplementary text). The designed LpPorA and DpPorA (Fig. 6a, b) were simulated three times for 200 ns each to investigate the stability of the constructed models, with one of the simulations extended to 500 ns (Supplementary Fig. 14). The stability of the pores is an important criterion for the validity of their structures. The RMSD calculations revealed a significant change in the RMSD values in each of the systems for the first 10–15 ns, following which the curve flattens and the graphs show a plateau. The structures obtained from the 200 ns equilibration runs were further employed in applied-field MD simulations. Since the diameters of the designed pores are in the order of Angstroms, several potassium and chloride ions can move through the pore simultaneously in either direction (Fig. 6a, b). Thus, we determined multi-ion 1D free energy surfaces (FESs) as a function of the channel axis *z* for K⁺ and Cl⁻ ions. For LpPorA, the K⁺ transport barrier is around 3.0 kcal/mol, whereas the Cl⁻ barrier is 3.5 kcal/mol (Fig. 6c). Compared with these values, the barriers for DpPorA are higher, with the barrier for Cl⁻ transport about 0.5 kcal/mol higher than that for K⁺ ions (Fig. 6d). In the next step, a transmembrane potential was applied in the simulations using the constant electric-field approach to calculate the ionic conductance values (Supplementary Table 2). The designed pores show good stability during the applied-field simulations, as seen from the RMSD curves (Supplementary Fig. 14). At an applied voltage of +0.5 V, the conductance was calculated to be ~3.68 nS for LpPorA and ~1.46 nS for DpPorA. In DpPorA, the C-terminal residues of two helices partially inhibit ion transport, which was not observed for LpPorA (Supplementary Fig. 15). These data indicate the distinct helical packing of L and DpPorA helices in the lipid bilayer associated with the stereo inversion of amino acids, which results in altered structural pore conformation in agreement

with single-channel studies. In addition, the ratio of the K⁺ and Cl⁻ current ($I_+/I_-$) obtained from MD simulations shows that both pPorA pores are cation-selective (Fig. 6c, d and Supplementary Fig. 15).

Interestingly, unbiased MD simulations indicated that the binding of cationic am₈γCD with pPorA was stabilized by strong hydrogen bonding and electrostatic interactions with the carboxyl groups of the aspartic acid residues (Supplementary Fig. 16). The permeation of the cationic am₈γCD through LpPorA and DpPorA was explored using steered MD (SMD) simulations to estimate the force required to enable the transport process. The SMD force profiles were generated as the function of the reaction coordinate z, i.e., the center of mass (COM) distance between am₈γCD and the $C_\alpha$ atoms of the pPorA, by pulling the ligand molecule along the channel axis of pPorA. In addition, the translocation of nonaarginine through DpPorA was analyzed by pulling the C-terminus arginine residue along the z-axis. Fig. 6e-h depicts snapshot structures of am₈γCD and nonaarginine inside the pore and the average forces required to pull the analytes through the pore. The average maximum force required to pull the cationic am₈γCD through DpPorA was ~400 pN and ~500 pN for nonaarginine through DpPorA, whereas the average maximum force required to pull the cationic am₈γCD through LpPorA was ~370 pN (Fig. 6e-h and Supplementary Fig. 16). The relatively larger force required to pull the am₈γCD through DpPorA compared to that of the LpPorA corroborate the greater binding affinity of am₈γCD with DpPorA, likely due to the slightly larger LpPorA pore diameter (Fig. 6e-h and Supplementary Fig. 16). It was observed that the maximum force necessary at the N-terminus entry point of the pore is due to strong electrostatic interactions between am₈γCD and the negatively charged glutamic acid (E3) and aspartic acid (D7) residues. Similarly, nonaarginine was significantly stabilized by the acidic amino acid residues and a large force was necessary to overcome the strong electrostatic interactions. Notably, there is a considerable difference in the mode of interaction

of am$_8$γCD and nonaarginine with the pore. The bulky spherical shape of am$_8$γCD is responsible for the strong interactions at the entry position around z = −20 Å (Fig. 6g). In contrast, the disordered structure of nonaarginine allows the C-terminus to permeate through the pore. However, after the permeation of the C-terminus, the remaining residues interact rather strongly with the acidic amino acid residues of the pore, reflected in the force profile around z = +10 Å (Fig. 6h).

MD simulations of the cWza pore and calculation of the RMSD values also suggested reasonable stability of the newly constructed pore after initial structural deviation (Supplementary Fig. 17). A 500 ns-long unbiased MD simulation performed for both L and DcWza strongly indicates the absence of structural changes after 200 ns. The multi-ion free energy barriers for LcWza were 2.0 kcal/mol and 3.0 kcal/mol for K$^+$ and Cl$^-$ ions, respectively. The barrier for K$^+$ ions through DcWza is 0.5 kcal/mol higher than LcWza, while the barrier for Cl$^-$ ions is the same (Supplementary Fig. 17). In this case, the K$^+$ ions must overcome two barriers located at −15 Å and 10 Å. The cationic residues R331 and R351 on both sides of the cWza repel the K$^+$ ions producing barriers at the two ends, whereas the anionic acidic residues and histidine in the middle at around −5 Å stabilize the K$^+$ ion. The width of the free energy barrier for Cl$^-$ at −5 Å is rather large due to the repulsion by acidic residues, and the calculated free energy surfaces suggest that all designed pores are cation-selective. For LcWza, applied-field simulations showed conductance values of -1.11 and -1.89 nS, whereas DcWza exhibited conductance values of -1.23 and -1.20 nS at voltages −0.5 V and +0.5 V (Supplementary Table 2). Structural analysis revealed that each cWza monomer is positively charged with four basic amino acid residues (three arginine amino acids and one lysine) and only two acidic residues (one aspartic and one glutamic acid). However, two arginine residues are oriented toward the interior of the channel, neutralizing the charges of the two acidic residues. Further, the cWza pore contains proline residues close to both ends of the channel, and the presence of asparagine and proline residues makes the C-terminal prone to a random coil conformation. We propose that the flexible tails frequently inhibit the ion transport in both L and DcWza pores and may be responsible for the low conductance values compared to those of the pPorA pores (Supplementary Fig. 18). Notably, attempts to employ cWza directly from the Wza crystal structure (PDB ID: 2J58) in simulations failed due to unreasonably high conductance values[39] (Supplementary Fig. 19).

## Discussion

In this study, we engineered alpha-helical pores comprising entirely D-amino acids based on two natural bacterial membrane pores[22,29,38,39]. The pore properties of DpPorA, specifically the single-channel conductance, was distinct from that of LpPorA. We suggest that stereo inversion of amino acids into dextrorotatory forms most likely alters the surface topology, helical packing and structural assembly of the pores leading to the lower ion conductance of DpPorA[25–28]. The radius profiles for LpPorA and DpPorA were calculated using HOLE (Fig. 7a)[45]. The minimal average radii for LpPorA and DpPorA are 4.97 and 4.02 Å, respectively. In a simplistic model assuming a cylindrical pore with radius r and length L, the conductance $G_p$ is estimated to be

$$G_p = \frac{k\pi r^2}{L} \tag{1}$$

where k denotes the conductivity of the electrolyte[46]. Identifying the radius of this model with the above-mentioned minimal radius, LpPorA should be 1.5 times more conductive than DpPorA. At the same time, it should be noted that the conductance is not solely influenced by the pore radius but also by the internal charge distribution. Thus, electrostatic potential maps were computed, as depicted in Fig. 7.

The electrostatic potential at the inner surfaces of L and DpPorA is negative. The potential at the C-terminal side (Fig. 7b–e) of LpPorA is

more negative than that of DpPorA, while the potential at the N-terminal side of DpPorA is more positive. The negative electrostatic potential at the C-terminal side is because of the interior acidic D28 and E31 residues and N21. Overall, it is evident that the variation in radius profile and electrostatic potential at the inner channel surface within the two pores is responsible for the observed conductance differences between DpPorA and LpPorA. Also, the similarity in the radii and electrostatic potential for DcWza and LcWza are in accordance with the similar conductance values of these two pores (Supplementary Fig. 20 and Supplementary text). Notably, both D and LpPorA self-assembled into stable octameric preoligomers, confirming their identical subunit compositions. Interestingly, these preoligomers directly insert into lipid bilayers yielding stable pores and do not require additional protein domains[22,29]. In contrast, DcWza pore formation occurs via the oligomerization of the monomers upon membrane binding. These peptide sequences based on natural porins that show highly contrasting pore formation behavior in the membrane environment highlight the challenges in designing alpha-helical sequences to obtain defined self-assembled structures.

Our data show that D-peptide pores are highly resistant to protease, whereas the L peptide pores are degraded and their ordered oligomeric structures destabilized. We propose that the conformation of the pore associated with specific side-chain interactions of D-amino acid within the helix most likely hinders the accessibility of the pore to the protease enzyme. The protease-resistant designed D-peptide pores might be effective as antimicrobial and anticancer agents, creating a new arena for developing better therapeutic agents[47–49]. Our future studies will focus on the in vivo application of these peptides in drug delivery systems for cancer therapy. Previous studies have demonstrated applications of nanopores in single-molecule sensing of low mass analytes, lengthened polymer chains, peptides and nucleic acid sequencing.[1–6,50,51] More recently, protein nanopores are emerging as promising candidates in proteomics for protein sequencing[52–56]. Our designed transmembrane pores composed of mirror-image peptides with defined structures add a new class of functional nanopores for single-molecule sensing of large cyclic sugars, polypeptides and polymers. These pores are autonomously assembled as preoligomers and this self-assembly permits relatively straightforward purification of the peptides compared to biological nanopores. We used peptide synthesis to build structurally stable pores from D-amino acids for selective sensing, which is challenging to achieve with natural membrane pores. Engineering such synthetic transmembrane pores can find applications in membrane protein folding design to develop sophisticated nanopores[57,58]. We emphasize that such designed pores will be advantageous for applications in nanobiotechnology for the characterization of complex biomacromolecules.

## Methods

### Single-channel electrical recordings

Electrical recordings were carried out by using bilayers of 1,2-diphytanoyl-$sn$-glycero-3-phosphocholine (DPhPC, Avanti Polar Lipids) formed across -100 μm in diameter polytetrafluoroethylene (Teflon) film of 25 μm thick (Goodfellow, Cambridge)[59]. The Teflon film is sandwiched between two sides of the Delrin bilayer chamber, called cis and trans compartments (500 μL each). Solvent-free bilayers were formed by pre-treating the aperture with hexadecane in n-pentane (1 μL, 5 mg mL$^{-1}$) on each side of the bilayer chamber. Then the bilayer chamber was filled with the electrolyte solution (1 M KCl, 10 mM HEPES, pH 7.4) and DPhPC in n-pentane (2 μL, 5 mg mL$^{-1}$) was added to both sides of the chamber. After 5 min, a bilayer was formed when the electrolyte was raised, bringing the two lipid surface monolayers together at the aperture. pPorA pores were formed by adding

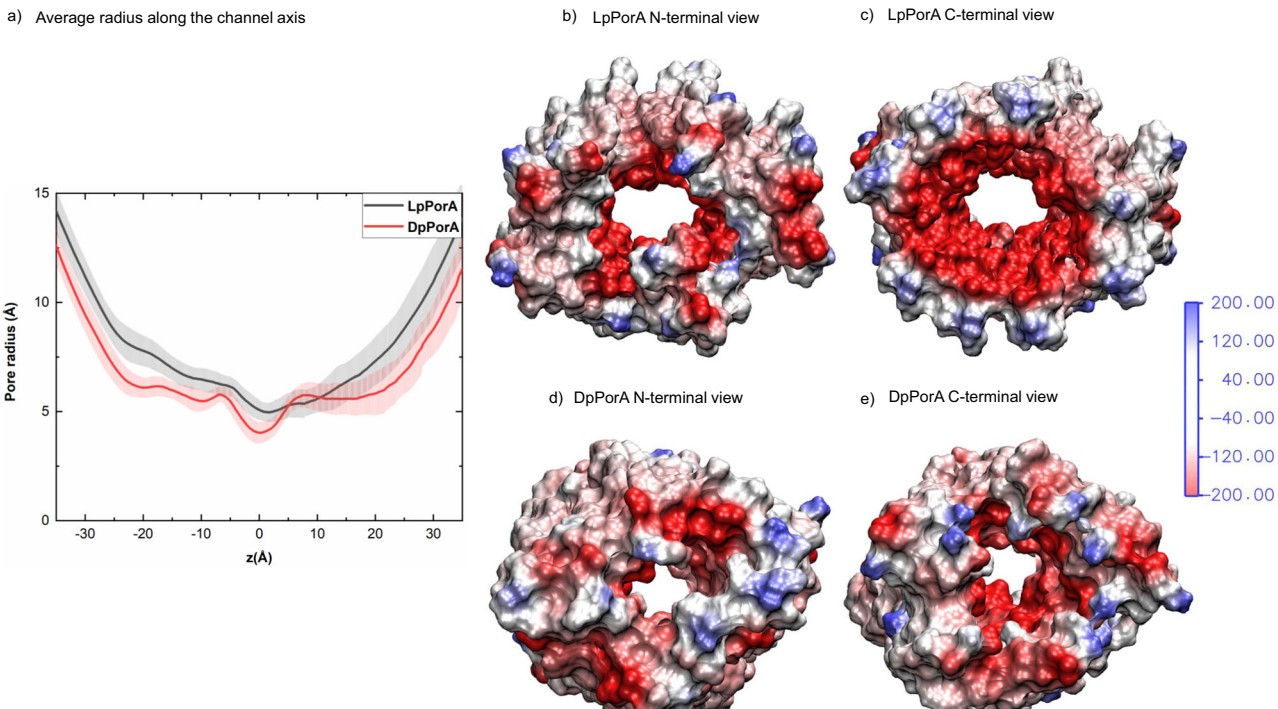

**Fig. 7 | Radius along the channel axis and electrostatic potential of DpPorA and LpPorA. a** Radius profile along the channel axis averaged over a trajectory also showing the standard deviations as shaded error bars. **b** N-terminal and **c** C-terminal view of the electrostatic potential map of the amino residues of LpPorA lining the interior channel wall. **d** N-terminal and **e** C-terminal view of the electrostatic potential map of the amino residues of DpPorA lining the interior channel wall. The computed electrostatic potential ranges from −580 to +186 $k_BT/e$ for LpPorA and −512 to +180 $k_BT/e$ for DpPorA, where 1 $k_BT/e$ = 26 mV at 300 K. For clarity, the color range for the electrostatic potential has been limited from −200 to +200 $k_BT/e$.

self-assembled preoligomers extracted from the SDS PAGE gel to the cis side of the chamber. The cWza pores were formed by adding peptide solution in 0.1% DDM (1 μL, 100 μg mL$^{-1}$) to the cis side. The cis chamber was connected to the grounded electrode and the trans chamber was attached to the working electrode. A potential difference was applied through a pair of Ag/AgCl electrodes, set in 3% agarose containing 3.0 M KCl. We investigated the ion selectivity of pores using the KCl salt gradient applied across the bilayer chambers (1 M cis /0.15 M trans). The reverse potential was determined to calculate the permeability ratio of K$^+$ and Cl$^-$ ions across the pore (Supplementary text). The current was amplified using an Axopatch 200B amplifier, digitized with a Digidata 1550B digitizer (pCLAMP 11.0, Molecular Devices, CA) with a low-pass filter frequency of 10 kHz and a sampling frequency of 50 kHz or a low-pass filter frequency of 2 kHz and a sampling frequency of 10 kHz. Single-channel data analysis was conducted using Clampfit 11.0 (Molecular devices, CA). Histogram plotting were performed in OriginPro 2022 (OriginLab).

### Transport across giant unilamellar vesicles

The gel-assisted swelling method was employed for vesicle preparation[60,61]. Briefly, DPhPC lipid film (30 μl, 1 mg/mL) was coated on polyvinyl alcohol or PVA (5% w/v) coated glass slides. The desiccated lipid film was then hydrated at room temperature with salt solution (0.1 M KCl, 10 mM HEPES pH 7), which promotes the formation of giant unilamellar vesicles. The vesicles were collected from the slide using a cut-tip. The DPhPC GUVs were doped with a few mol% of fluorescently labeled lipids (0.05 mol% ATTO-550 DOPE) for ease of microscopic imaging. The GUV solution was mixed with LpPorA/DpPorA (1 to 4 μM in 0.1% DDM) or DcWza protein (8 to 25 μM in 0.1 % DDM) and the fluorescent hydrophilic molecules to investigate the transport properties of peptides. Vesicle-protein solution was mixed with Alexa-Fluor 350 and

then added to reagent barrier chambers prepared on BSA passivated glass slides. Time-lapse imaging was done every minute for 10 min and then every five minutes for 30 min for a single vesicle or multiple vesicles in a single frame. After 30 min of time-lapse imaging, different individual vesicles were scanned and images were taken for statistical analysis. The same procedure was repeated for control vesicles with the same amount of detergent but without peptides. Detailed analysis of fluorescence intensity in vesicles is provided in the Supplementary text.

### Pore designing and MD simulations

First, LpPorA and LcWza were designed from the sequences using the CCBuilder web server[62]. The octameric structure of the LpPorA and LcWza were built from the sequence by varying three simple parameters – the pitch or pitch angle, the radius of the assembly, and the interface angle. The pPorA pore was designed with a pitch angle of 226 degrees, a 14 Å radius, and a 205-degree interface angle. Similarly, the cWza pore was designed with a pitch angle of 125 degrees, a 16.5 Å radius, and an interface angle of 120 degrees. The final pore radius of pPorA was 14 Å, whereas the pore radius of the cWza pore was 17 Å. DpPorA and DcWza were modeled using the BIOVIA Discovery Studio software, which converted the L-isomeric structure to the D-isomeric structure by switching the handedness of all chiral centers[63]. These starting structures were then subjected to unbiased MD simulation. The pores show reasonable stability and acceptable calculated conductance values, highlighting the experimental observations. The newly constructed channels were employed in unbiased and applied-field MD simulations. Using the CHARMM-GUI Membrane Builder[64], all designed pores were embedded in a 1,2-diphytanoyl-sn-glycero-3-phosphocholine (DPhPC) bilayer solvated with TIP3P water molecules on both sides of the membrane, maintaining a water thickness of 25 Å on each side of the membrane. All systems composed of the protein, membrane, solvent, ions, and ligand have roughly 85000 atoms. The

MD simulations were carried out with the CHARMM36-m[65] force field using the GROMACS[66] molecular dynamics software, version 5.1.4. The structures of the unbiased MD simulation after 200 ns were further used to determine the ion conductance in the presence of applied fields (Supplementary Text). The Ramachandran plot of the modeled pores was calculated, with details provided in the supporting information (Supplementary Text and Supplementary Fig. 21). Steered MD (SMD) simulations were performed by pulling the center of mass of am$_8$γCD and the C-terminus arginine residue of the nonaarginine from the N-terminal side of the pore toward the C-terminal side along the channel axis[67]. The SMD force profiles were calculated as a function of the reaction coordinate z. This reaction coordinate is defined as the COM distance between analytes and the Cα atoms of DpPorA. The SMD simulations were repeated three times with each analyte with a constant velocity of 1 Å/ns using a spring constant of 100 kJ/mol/nm$^2$.

## Data availability

All relevant data supporting the key findings of this study are available within the article and the Supplementary Information file. All raw data generated during the current study are available from the corresponding author upon request. Source data underlying Figs.1b, 3a, 3c and 5c are provided as a Source Data file. Source data are provided with this paper.

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

## Acknowledgements

This work was supported by the research grant awarded by the Department of Biotechnology, Government of India (BT/PR34466/BRB/10/1830/2019). KRM acknowledges the research grant of Science & Engineering Research Board (SERB), Department of Science and Technology, Government of India (CRG/2021/000622). K.R.M. thanks RGCB for intramural research funding support. H.B. thank 'Innovative Young Biotechnologist Award' of the Department of Biotechnology, Government of India (BT/11/IYBA/2018/09). H.B. thanks INSPIRE Faculty award, Department of Science and Technology, Government of India (DST/INSPIRE/04/2020/000015). S.K.R. is supported by Senior Research Fellowship from the Indian Council of Medical Research (5/3/8/9/ITR-F/2020). K.J. is thankful to the Alexander von Humboldt (AvH) foundation for an AvH postdoctoral research fellowship. K.J. and U.K. acknowledge the North German Supercomputing Alliance (Norddeutscher Verbund für Hoch- und Höchstleistungsrechnen – HLRN) for providing access to their high-performance computational facility. K.J. is grateful to Dr. Jigneshkumar Dahyabhai Prajapati for helpful scientific discussions.

## Author contributions

S.K.R. performed and analyzed current recordings. S.K.R., A.S., A.D. and D.V. determined the biophysical properties of peptides and analyzed the single-channel data. K.N. performed and analyzed fluorescence imaging of giant vesicles supervised by H.B., K.J. performed molecular dynamics simulations supervised by U.K., S.K.R. and K.R.M. conceived the study and designed experiments. All authors wrote and approved the paper. K.R.M. supervised the study.

## Competing interests

The authors declare no competing interests.
