## [Peer Review File · Nature Communications]

Assembly of Transmembrane Pores from Mirror-Image PeptidesReviewers' comments:

Reviewer #1 (Remarks to the Author):

The manuscript entitled "Assembly of Transmembrane Pores from Mirror-Image Peptides" authored by Smrithi Krishnan R. et al. has reconstituted a pore formed by D-type amino acid and characterized its membrane activity. However, this manuscript is not suitable to publish in Nature Communications with the following concerns:

The authors addressed 'The DpPorA peptides containing D-amino acids based on the natural membrane pore PorACj were produced using solid-phase peptide synthesis'. Then they used mass spectrometry to determine the outcome. A commonly used MS should not be capable to differentiate between chiral isomers. In addition, each peptide contains 40 AA, the synthesis process should introduce error rate, how the authors purify the pore forming peptides? Likely, the self-assemble process should produce heterogeneous oligomers (Each monomer has different artifacts at different site). Both MS and SDS-page are insufficient to characterize amino acid analogues. The synthesis of the 44mer peptide all contains unnatural amino acid with 100% efficiency does not exist.

The MD simulation of mirror-image peptide formed pores looks interesting. The related work based on BIOVIA Discovery Studio software switches the handedness of all chiral centers, while simple calculation of the stereo-mirror structure does not give sufficient demonstration on the protein actual structure. For example, how to calculate the chiral amino acid induced backbone relocating? Otherwise, simply flip the secondary structure does not help to understand the complex chiral interaction. Moreover, the authors did not show the analyte inside pore with respect of in silico analysis.

The authors performed GUV swelling assay to prove the dye molecule transport over time elapse. However, the transport of this dye molecule has no relationship with the molecule tested elsewhere in this manuscript. In addition, the authors did not show the electrophysiology results to prove the interaction of the molecule with pores, and the transport rate from trans side. Negative control processes are missing: wash-off step, empty GUV uptake comparison.

Other Concerns:

1. Maybe the authors need to state why this mirror-peptide is designed? What is the goal to modify all the chirality.
2. "Shed light on the mechanism of action of antimicrobial peptides" stated by the authors does not give any useful statement. Do the authors want to design a pore-forming channel targeting to microbial? Else there is no chiral protein channel for drug permeation.
3. Different salt concentration induce varies forces to maintain the α -helical structure. Have the authors tried higher salt concentration? Maybe an L-type peptide with critical sensing region modified into its chiral isomers would make the work more reasonable.
4. The authors can explain the reason why unidirectional insertion is observed? Protein reconstitution normally inevitably meets 'wrong' orientation insertion.

Therefore, this manuscript is too preliminary to publish in Nature Communications. The manuscript needs to be greatly improved before submitting to a more specialized Journal.

Reviewer #2 (Remarks to the Author):

Major points:

- In fig. 1, Single DpPorA insertion (S) at +100 113 mV and corresponding unitary conductance histogram by fitting the distribution to a Gaussian; however, multiple spikes are seen.
- In Fig. 2, b, no interaction with g-CD, and authors explain this based on the neutral state of g-CD. However, in other previous work it has been clearly indicated this not to be the case, see among others:

Gu LQ et al, Reversal of charge selectivity in transmembrane protein pores by using noncovalent molecular adapters. Proc Natl Acad Sci U S A. 2000 Apr 11;97(8):3959–3964

Asandei A et al, The Kinetics of Ampicillin Complexation by γ -Cyclodextrins. A Single Molecule Approach, et al., J. Phys. Chem. B, dx.doi.org/10.1021/jp204640t

- In fig. 2, c, experiments with peptides, the original traces look quite noisy, yet authors present a single closed substate. Some zoomed-in view and analysis of such traces would be needed to further substantiate such claims. No data is shown about the association kinetics of such peptides to the nanopore, it would be extremely useful in order to better shed light on the physical properties of the nanopore itself for analytes capture.

- In Fig. 4, a, b, authors show and analyze a single Gaussian peak, despite the clear-cut view of multiple transitions and spikes on the original traces.

- On line 394, authors state that ‘.The minimal average radii are 4.97 and 4.02 Å for LpPorA and DpPorA..’. Yet, from the analysis of data shown in Fig. 2, c, authors conclude ‘..an increasing voltage confirmed the translocation of these cationic peptides through the pore..’. Considering that the peptide diameter is most likely larger than nanopores diameter, how can the assertion stand correct? A more in-depth analysis would be required. Also, I recommend authors to present the voltage-dependent analysis in detail.

Minor points

- No description on how the nanopores selectivity was measured.

- There is plenty of previous work regarding the particular aspects of peptide-nanopore interactions; it would be very helpful for the interested readers a more comprehensive citation of such work and concise analysis and discussion of authors data vs. what’s already established in this field, that could indeed lead to ‘..new functional nanopore biosensors..’.

Reviewer #3 (Remarks to the Author):

This manuscript is a breath of fresh air and I have read it with great interest. The Authors introduce the importance of stereochemistry to the structure and function of nanopores, which I consider a very important contribution. Peptide synthesis was used to create pores using D-amino acids that were characterized along with nanopores composed of L-amino acids. The Authors designed a set of clever experiments that demonstrated conductance and selectivity of both types of nanopores as well as application of the pores in resistive-pulse experiments. The measurements together with the MD simulations allowed them to unravel distribution of charged amino acids along the pore length and how this distribution facilitates detection of cationic species. Clear differences in the conductance of L and D pores were revealed that were explained by MD modeling. Finally, a striking difference was demonstrated in the stability of these nanopores to protease reactions.

This manuscript certainly belongs to Nature Comm. and is bound to be cited a lot. I am convinced the nanopore structures the Authors presented will be used in fundamental studies as well as biotechnology. It is a very comprehensive work.

I have only one minor comment – caption to Figure 1 does not provide all information; especially it is not clear which concentration gradient was used to obtain data in panel (d).

We would like to thank all three reviewers for their thoughtful comments and suggestions. We have addressed each comment and made substantial changes that have significantly improved the quality of the revised manuscript.

Reviewer 1 Comments:

1. The authors addressed 'The DpPorA peptides containing D-amino acids based on the natural membrane pore PorACj were produced using solid-phase peptide synthesis'. Then they used mass spectrometry to determine the outcome. A commonly used MS should not be capable to differentiate between chiral isomers.

The synthesized peptides were purified by HPLC, and their mass was evaluated using mass spectroscopy. Subsequently, we recorded the circular dichroism (CD) spectrum of the synthesized DpPorA peptide and compared it with the corresponding LpPorA peptide. The equivalent but opposite ellipticities observed in the CD spectrum of the two peptides suggest that the peptides were indeed chiral isomers. We completely agree with the reviewer that a commonly used MS cannot differentiate between chiral isomers. Here, we have used MS only to evaluate the mass of synthesized peptides. In the submitted manuscript, we have stated this "*their mass was evaluated by mass spectrometry*"). We have clarified this in the revised manuscript (main text page 5, Figure 1 and Figure S1). In the revised manuscript, we have also provided detailed synthesis and purification procedures of the peptides, including HPLC traces (SI text and Figure S1).

2. In addition, each peptide contains 40 AA, the synthesis process should introduce error rate, how the authors purify the pore forming peptides? Likely, the self-assemble process should produce heterogeneous oligomers (Each monomer has different artifacts at different site). Both MS and SDS-page are insufficient to characterize amino acid analogues. The synthesise of the 44mer peptide all contains unnatural amino acid with 100% efficiency does not exist.

We thank the reviewer for bringing this to our attention. It is important to note that the monomeric DpPorA peptides spontaneously self-assemble to form octameric pre-oligomers that remain stable in SDS PAGE. When the gel band corresponding to the preoligomers was extracted and added to a planar lipid bilayer, we observed consistent, stable pore formation ($n = 100$) in single-channel recordings. We emphasize that this property of synthetic DpPorA to autonomously assemble to form stable preoligomers and large transmembrane uniform pores is noteworthy and unique. Importantly, the formation of SDS stable preoligomers is rare compared to other pore-forming peptides reported in the literature (Mahendran, K.R. et al. Nat. Chem. 2017. 9, 411-419 and Scott, A.J. et al. Nat. Chem. 2021. 13, 643-650). This self-assembly also permits relatively straightforward purification of the peptide preoligomers compared to biological proteins.

As the reviewer has correctly stated, a small percentage of the peptide monomers may contain artifacts. Peptides containing such random artifacts are unlikely to participate in the formation of such stable assemblies. Therefore, a second level of purification takes place by extracting the preoligomer band from the SDS PAGE gel. Since the single-channel recordings were carried out using the preoligomers extracted from the gel, the effect of any artifacts on single-channel recording data can therefore be considered negligible. Importantly, our data is based on three separate peptide preparations, wherein the preoligomerization and stable pore formation (100 single-channel events) remain consistent, as demonstrated by SDS PAGE and single-channel recordings. In each case, the observed pores are functional, as shown by the binding of cationic CD and polypeptides. We have discussed this in the revised manuscript (see main text page 4-10, Figure 1, Figure 2, SI text, Figure S2, Figure S3, Figure S5 and Figure S7).

Figure R1: Electrical and functional properties of DpPorA

a) SDS-PAGE gel showing the DpPorA preoligomer. b) The mean unitary conductance histogram of DpPorA at +100 mV. c) DpPorA pore at +50 mV. d) DpPorA with 10 μM am₈γCD at -50 mV (cis) and +50 mV (trans). e) DpPorA with 10 μM R9 at -50 mV (cis) and +50 mV (trans).

The pore-formation ability of the peptide preoligomers is re-affirmed by our additional experiments to evaluate the stability of the synthesized DpPorA to protease degradation. After Proteinase K treatment, the DpPorA remained stable and formed functional pores. In contrast, the LpPorA were degraded and did not form functional pores after protease treatment (main text page 10-12, Figure 3, SI text and Figure S8).

3. The MD simulation of mirror-image peptide formed pores looks interesting. The related work based on BIOVIA Discovery Studio software switches the handedness of all chiral centers, while simple calculation of the stereo-mirror structure does not give sufficient demonstration on the protein actual structure. For example, how to calculate the chiral amino acid induced backbone relocating? Otherwise, simply flip the secondary structure does not help to understand the complex chiral interaction.

In the MD simulations, the L-amino acids have been replaced by their D-amino acid counterparts. As written in the methods section on page 25: “DpPorA and DcWza were modeled using the BIOVIA Discovery Studio software, which converted the L-isomeric structure to the D-isomeric structure by switching the handedness of all chiral centers.” BIOVIA Discovery studio is able to accurately convert an L-amino acid to its D-amino acid isomeric configuration. In the present case, the whole LpPorA pore was converted into the DpPorA pore. To this end, the BIOVIA Discovery studio software not only converted all L-amino acids into D-amino acids configuration but also changed the left-handed helices to right-handed ones. **Figure R2** shows that the converted LpPorA and DpPorA pores are mirror images of each other. An analysis demonstrating that most of the altered D-amino acids reside in the left-handed helix region of the Ramachandran plot (new data) is shown in the revised manuscript. We have included and clarified these points in the revised manuscript (main text page 16-20, page 25, Figure 6, SI text, Figure S13, S14, S16, S17 and S20).

Figure R2: The L and D pores converted using BIOVIA discovery studio. a) Superimposed structures of LpPorA and DpPorA, which are mirror images of each other. b) Superimposed structure of LcWza and DcWza. The green cartoon structures represent the L-pores, whereas the magenta line-ribbon represents the D pores.

Subsequently, long MD simulations were performed to find the equilibrium conformation of the respective pores. Due to the different chiralities of the involved amino acids, LpPorA

and DpPorA evolved into different pore structures, for which representative conformations are shown in Figures 6 and 7 of the main manuscript. Thus, the relocation of the backbone is automatically determined during these unbiased MD simulations. While it is not efficient to display the relation of individual backbone atoms, we believe that the differences in the overall structure of LpPorA and DpPorA are clearly visible (Figures 6 and 7). Moreover, structural changes were observed in the radii of the L- and D-pores, the distributions of electrostatic potential maps, and multi-ion free energy profiles (main text page 20-21, Figure 6, Figure 7 and SI text).

4. Moreover, the authors did not show the analyte inside pore with respect of in silico analysis.

We performed MD simulations with the cationic am₈γCD, which was presented in Figure S15 of the original submitted manuscript. However, Reviewer 1 likely has overlooked the representative snapshots of LpPorA and DpPorA with the cationic am₈γCD. As reviewer 1 recognized the importance of this data, steered MD (SMD) simulations were carried out to demonstrate the permeation of the cationic am₈γCD and nonaarginine through DpPorA. We have included this new data in the updated Figure 6 and discussed this in the revised manuscript (main text page 18-19, Figure 6, Figure S15 and SI text).

Steered MD simulations to show the molecular transport of charged peptides and cationic CDs to highlight the charge-selectivity of the pore

The permeation of the am₈γCD through the DpPorA pore was explored using steered MD simulations (SMD) to estimate the force required to enable the transport process. The SMD force profiles were generated as the function of the reaction coordinate z , i.e., the center of mass (COM) distance between am₈γCD and the C α atoms of the DpPorA, by pulling the ligand molecule along the channel axis of DpPorA. The translocation of nonaarginine through the DpPorA pore was analyzed by pulling the C-terminus arginine residue along the z -axis. **Figure R3** depicts snapshot structures of am₈γCD and nonaarginine inside the pore and the average forces required to pull the analytes through the pore. The average maximum force required to pull am₈γCD was ~400 pN, while that for nonaarginine was ~ 500 pN. It has been observed that the maximum force necessary at the N-terminus entry point of the pore is due to strong electrostatic interactions between the am₈γCD and the negatively charged glutamic (E3) and aspartic acids (D7). Similarly, nonaarginine was also significantly stabilized by the acidic amino acid residues and a large force was necessary to rupture the strong electrostatic interactions. Notably, the detailed

interactions with the pore surface are quite different for am₈γCD and nonaarginine. The bulky spherical shape of am₈γCD is responsible for the strong interactions at the entry position around z=-20 Å. In contrast, the disordered structure of nonaarginine allows the C-terminus to permeate through the pore. Soon after that point, however, the remainder of the nonaarginine interacts rather strongly with the pore's acidic amino acid residues, which is reflected in the force profile at around z=+10Å. We have included this new data in the main figure 6 and main text page 18 and 19.

Figure R3: a) Transport of am₈γCD and nonaarginine through DpPorA b) Average force profile from steered MD simulation for the permeation of am₈γCD and nonaarginine as the function of reaction coordinate z, i.e., the COM distance between the analytes and the Ca atoms of DpPorA.

5. The authors performed GUV swelling assay to prove the dye molecule transport over time elapse. However, the transport of this dye molecule has no relationship with the molecule tested elsewhere in this manuscript. In addition, the authors did not show the electrophysiology results to prove the interaction of the molecule with pores, and the transport rate from trans side. Negative control processes are missing: wash-off step, empty GUV uptake comparison.

We thank the reviewer for bringing this to our attention which significantly improved the quality of the manuscript.

✓ The fluorescent molecule Alexa-Fluor 350 hydrazide is a hydrophilic molecule that does not traverse the membrane without the presence of pore-forming proteins. Indeed, after reconstitution of DpPorA in membranes and the addition of the molecule outside the vesicles, real-time uptake of the molecule is observed across the vesicles. The reviewer has pointed correctly that there is no relationship with the molecules tested elsewhere in the manuscript. This assay was conducted to confirm the pore-forming activity of DpPorA in alternative membrane models apart from the free-standing membranes used in single-channel recordings (**Figure R4**). Such assays employing GUVs have been employed to understand the pore-forming properties of

membrane proteins (Cama et al. J. Am. Chem. Soc. 137, 13836–13843, 2015, Dezi et al. Proc. Natl. Acad. Sci. U. S. A. 110, 7276–7281, 2013, Barba-Bon et al. J. Am. Chem. Soc. 141, 20137–20145, 2019). **We emphasize that the GUV transport assay performed here is an alternate membrane model system to confirm the pore formation of DpPorA.** We have created a separate section with new figure 4 for transport across GUVs to clarify this in the revised manuscript (main text page 12-14, new Figure 4 and Figure S9).

Figure R4: Functional assembly of DpPorA in giant vesicles

a) Schematic and fluorescence image of single vesicle outlining the transport of molecules in the absence of pore. b) Schematic and fluorescence image of single vesicle revealing dye transport over time in the presence of DpPorA, where false blue color represents the Alexa-Fluor 350 dye in vesicles, inset shows the phase-contrast image of the vesicle. c) Normalized intensity of vesicles with DpPorA and without peptide is represented over time. d) Vesicle permeabilization rate is shown in the absence and presence of DpPorA and control. Buffer conditions: 0.1 M KCl, 10 mM HEPES pH 7; scale bar: 10 μ m.

- ✓ Negative control processes- The wash-off step is not necessary for the experiments here as we are not measuring the binding of molecules on membranes or release of molecules from inside but measuring the uptake of Alexa-Fluor 350 molecule across DpPorA reconstituted in membranes via diffusion from outside the vesicle to inside.

- ✓ Empty GUV uptake- This is a rather important control mentioned by the reviewer and this data has already been included in the supplementary information (Figure S9 and main figure 4b). The control vesicles without DpPorA reconstituted do not transport any Alexa Fluor 350 across their membranes. To make the negative control data clearer to the reader, we have added the microscopic image of the control/empty GUV with the molecule signifying no transport across empty vesicle (see main Figure 4).
- ✓ We have also included the new data of LpPorA pore formation in the vesicle system (see main text 14 and Figure S9).

6. Maybe the authors need to state why this mirror-peptide is designed? What is the goal to modify all the chirality?

Transmembrane pores based on α -helices have remained relatively unexplored and are emerging as a hot topic in nanobiotechnology and synthetic chemical biology (Mravic et al. Science 2019, Xu et al. Nature 2020 and Scott et al. Nature Chemistry 2021). Our motivation was to design and engineer simplified transmembrane alpha-helical pores that conduct specific ions and molecules. Here, we show that unnatural amino acids can be incorporated by chemical synthesis into peptides to build stable transmembrane pores. The mirror peptide was specifically designed to build a functional alpha-helical pore with improved biostability. We suggest that the designed D-peptide pores might be effective as antimicrobial and anticancer agents because of their stability to proteases. Accordingly, we evaluated the stability of the synthesized D-peptide to protease degradation. After Proteinase K treatment, the D peptides remained stable and formed functional pores. In contrast, the L-peptides were degraded. We emphasize that such membrane-spanning α -helix-based assemblies would add new structural motifs to the field. We conducted this work to pave the way for new selective nanopore sensors. We have expanded our abstract, introduction and discussion to highlight this (main text page 2-4, 22 and 23).

To demonstrate the structural and functional versatility of the pore, we have included additional experimental data in the revised manuscript.

- ✓ **DpPorA as a nanoreactor to follow covalent chemistry at the single-molecule level.**

We demonstrate the use of DpPorA pores as nanoreactors to investigate single-molecule covalent chemistry. For example, cysteines at the 24th position, placed in the ion conduction pathway, can act as sites for site-specific chemical modification using activated PEG thiol blockers (**Figure R5**). The addition of 1 mM monomethoxy poly (ethylene glycol)-*o*-pyridyl disulfide) (MePEG-OPSS-1k) to the cis side of the pore produced a step-wise closure of the pore at +50 mV, indicating a covalent modification of the cysteines (disulfide bond formation) (**Figure R5**). The pore was reverted to its open conductance state on the addition of 10 mM dithiothreitol (DTT) as disulfide bonds are cleaved, demonstrating a high specificity of the covalent modification (**Figure R5**). We have included this new data in the main figure 2, main text page 10 and Figure S6.

Figure R5: Reversible chemical modification of DpPorA with PEG polymers
a) Schematic pore model and reaction scheme showing the reversible interaction of DpPorA with MePEG-OPSS-1K. b) Electrical recording of MePEG-OPSS-1K mediated pore closure events at +50 mV and DTT mediated opening of the blocked pore at +50 mV.

6. “Shed light on the mechanism of action of antimicrobial peptides” stated by the authors does not give any useful statement. Do the authors want to design a pore-forming channel targeting to microbial? Else there is no chiral protein channel for drug permeation.

As stated in our previous response, we propose DpPorA as a possible therapeutic target as an antimicrobial agent due to its increased stability to natural protease enzymes. We also propose that apart from the known mechanisms exhibited by antimicrobial peptides (AMPs), this synthetic D-peptide could be a new class of AMPs that can interrupt host membrane integrity by forming stable pores with better stability to natural protease activity.

We included the quoted sentence to state that our results may further the understanding of interactions between pore-forming AMPs and lipid membranes, a step toward the optimized development of peptides for clinical purposes. We have modified this statement to ensure this is clear in the discussion of the revised manuscript (main text page 22).

7. Different salt concentration induce varies forces to maintain the α -helical structure. Have the authors tried higher salt concentration? Maybe an L-type peptide with critical sensing region modified into its chiral isomers would make the work more reasonable.

We thank the reviewer for this. As suggested by the reviewer, we have examined the effect of salt concentration (0.15 M KCl, 1M KCl and 3M KCl) on the single-channel properties of the DpPorA pore reconstituted into planar lipid bilayers (**Figure R6**). We observed that the conductance is a function of the salt concentration and the pore remained in the stable open state at all studied salt conditions. This indicated the stable alpha-helical pore structures at different salt conditions and the voltage-dependent gating property of the pore remained the same irrespective of different salt conditions. We have included this new data and discussed it in the revised manuscript (main text page 6,7 and Figure S4).

Figure R6: Single-channel properties of DpPorA at different electrolyte buffer concentrations. a) DpPorA in 0.15M KCl (+50 and +75 mV). b) DpPorA in 1M KCl (+50 and +75 mV). c) DpPorA in 3M KCl (+50 and +75 mV).

Please note that we performed charge selectivity measurements at different salt concentration gradients. We have included a detailed description in the revised manuscript (see SI text, main Figure 1 and main text page 6, 24).

Modifying the L-peptide sensing region into its chiral isomers is interesting. However, the modification of crucial residues may disrupt the autonomous self-assembly and such peptides may remain susceptible to protease degradation. In the future, we will work on this idea. We thank the reviewer for this suggestion.

8. The authors can explain the reason why unidirectional insertion is observed? Protein reconstitution normally inevitably meets 'wrong' orientation insertion.

Here we identified the orientation of DpPorA based on the interaction with the analyte, cationic $\alpha\text{M}_8\gamma\text{CD}$. The addition of $\alpha\text{M}_8\gamma\text{CD}$ to the cis side resulted in ion current blockades. Specifically, $\alpha\text{M}_8\gamma\text{CD}$ blocked the pore rapidly, indicating strong electrostatic interaction. Next, we perfused the CDs added to the cis side and then added $\alpha\text{M}_8\gamma\text{CD}$ to the trans side of the same pore so that the same orientation of the pore was maintained. Notably, $\alpha\text{M}_8\gamma\text{CD}$ from the trans side produced ion current blockades distinct from the cis side addition. Based on statistical analysis (25 independent single-channel experiments), we observe that $\alpha\text{M}_8\gamma\text{CD}$ bind strongly to the cis side of DpPorA, resulting in rapid pore closure compared to the trans side in 95% of experiments ($n = 24$ experiments). This asymmetry in CD binding indicated the possible preferred unidirectional orientation of the pore in the lipid membrane (**Figure R7**). We suggest that the cis side of the pore consists of abundant negatively charged residues as opposed to the trans side of the pore leading to the relatively stronger electrostatic CD binding. In the revised manuscript, we have included a schematic of the pore orientation based on the CD binding kinetics and a bar graph to clarify this (see main text page 7-8, main figure 2 and Figure S6).

We thank the reviewer for their brilliant observation and suggestion. We suggest that the presence of tryptophan in the terminal position might play a role in the unidirectional orientation, as shown previously for cWza peptide pores (Mahendran, K.R. et al. Nat. Chem. 2017. 9, 411-419). We have discussed this in the revised manuscript citing relevant references (main text page 22).

Figure R7: Orientation of DpPorA in the lipid membrane. Asymmetry in the cationic cyclic sugar binding with DpPorA and schematic showing orientation of DpPorA in lipid bilayers based on the cationic am₈γCD binding. A bar graph representing the pore orientation based on the CD binding is shown below.

Reviewer 2 Comments. Major points:

1. In fig. 1, Single DpPorA insertion (S) at +100 mV and corresponding unitary conductance histogram by fitting the distribution to a Gaussian; however, multiple spikes are seen.

We thank the reviewer for bringing this to our attention. The given mean unitary conductance histogram has been plotted based on 50 independent single-channel insertion events with a sharp peak of ~1.5 nS at +100 mV that revealed the homogeneity of the pores. We have added a representative current trace of a single DpPorA insertion at +100 mV that showed a conductance of ~1.5 nS (**Figure R8**).

The reviewer is correct that we have observed multiple spikes, which are gating events at observed specifically high voltages (+100 mV). In the revised manuscript, we have added the current amplitude histogram for DpPorA insertion at +100 mV to explain the multiple spikes. We have also added single DpPorA insertion and corresponding current amplitude histogram at +50 mV where the multiple spikes were not observed. We have modified Figure 1 accordingly and clarified this in the revised manuscript (main text page 6 and Figure 1 and Figure S2)

Figure R8: Electrical recordings of DpPorA (S). Single DpPorA insertion (S) at +50 mV and current amplitude histogram. Single DpPorA insertion (S) at +100 mV with multiple spikes and current amplitude histogram.

2. In Fig. 2, b, no interaction with g-CD, and authors explain this based on the neutral state of g-CD. However, in other previous work it has been clearly indicated this not to be the case, see among others:

Gu LQ et al, Reversal of charge selectivity in transmembrane protein pores by using noncovalent molecular adapters. Proc Natl Acad Sci U S A. 2000 Apr 11;97(8):3959–3964
 Asandei A et al, The Kinetics of Ampicillin Complexation by γ -Cyclodextrins. A Single Molecule Approach, et al., J. Phys. Chem. B, dx.doi.org/10.1021/jp204640t

- ✓ We thank the reviewer for pointing out this. The work cited by the reviewer has been carried out in alpha hemolysin and is impressive. Specifically, the transport of neutral beta and gamma cyclodextrin through this pore has been demonstrated.
- ✓ Here, we investigated the binding of cyclodextrins through the cation-selective peptide pore. We highlight that the pore lumen is lined by negatively charged residues that act as an affinity site facilitating the selective binding of cationic α -CD. The addition of α -CD to the cis side resulted in ionic current blockages at negative voltages as CDs are electrophoretically driven through the pore, promoting electrostatic interactions with negatively charged residues. Here applied voltage serves as a driving force facilitating the pulling and binding of the cationic CDs. No blockages were observed at positive voltages suggesting electrostatic repulsion of CDs. Next, the addition of α -CD to the trans side resulted in current blockages only at positive voltages in agreement with electrophoretic binding of the CDs. Based on this data, we propose that the negative charge pattern inside the pore lumen was exploited to develop a cation-selective nanopore sensor (**Figure R9**). In the revised manuscript, we have included a schematic of the electrophoretic binding of cationic CDs with the pore surface and current traces to clarify this (main text page 7-9, Figure 2 and Figure S5).

Figure R9. Interaction of DpPorA with cationic $\text{am}_{8\gamma}\text{CD}$
 a) Schematic and voltage-driven electrophoretic translocation of $\text{am}_{8\gamma}\text{CD}$ through DpPorA (10 μM , cis) at -75 mV. b) Schematic and electrophoretic repulsion of $\text{am}_{8\gamma}\text{CD}$ from DpPorA (10 μM , cis) at +75 mV c) Schematic and voltage-driven electrophoretic translocation of $\text{am}_{8\gamma}\text{CD}$ through DpPorA (10 μM , trans) at +75 mV. d) Schematic and electrophoretic repulsion of $\text{am}_{8\gamma}\text{CD}$ from DpPorA (10 μM , trans) at -75 mV. The inset shows the corresponding current-amplitude histogram.

- ✓ To confirm the charge-selective binding of the cationic CDs, we added neutral γ CD that did not produce any ion current blockages with the nanopore at different experimental conditions consistent with the cationic-selective nature of the pore. We attribute the different binding patterns of neutral γ CD with alpha hemolysin and peptide nanopore to the distinct charge pattern of the pore. Notably, the molecular dynamics simulations highlighted the electrostatic potential map of the amino residues of the peptide nanopore lining the interior pore wall. We have explained and clarified this in the revised manuscript (main text 9, Figure 2, Figure 7 and Figure S6)
- ✓ We also examined the interaction on anionic $s_8\gamma$ CD with the pore. No interaction was observed at different experimental conditions, indicating high selectivity and specificity of the pore to cationic molecules. We have included new data to explain this in the main manuscript (main text 9, Figure 2 and Figure S6).

3. In fig. 2, c, experiments with peptides, the original traces look quite noisy, yet authors present a single closed substate. Some zoomed-in view and analysis of such traces would be needed to further substantiate such claims. No data is shown about the association kinetics of such peptides to the nanopore, it would be extremely useful in order to better shed light on the physical properties of the nanopore itself for analytes capture.

- ✓ We thank the reviewer for the excellent suggestion. As suggested, we added zoomed-in ion current traces to precisely show the time-resolved ion current blockages induced by nonaarginine (R9) (**Figure R10**). We show that cationic R9 are electrophoretically driven through the pore, producing ion current blockages with applied voltage serving as a driving force facilitating the electrostatic interaction of peptides. We have modified the figure in the revised manuscript (main figure 2 and Figure S7).
- ✓ Pore blocking by the R9 was quantified using a statistical analysis of the pore in its unblocked and 'blocked states. The kinetic rate constants such as the association rate (k_{on} , rate of R9 entry), the dissociation rate (k_{off} , rate of R9 exit) and the equilibrium binding constant (K_D) was determined. Accordingly, we quantified the voltage-dependent translocation kinetics of R9 peptides through DpPorA. The mean residence time of peptide blocking (τ_{off}) and time between successive blocking (τ_{on}) was determined. The dissociation rate ($k_{off} = 1/\tau_{off}$) increased with increasing voltages, indicating translocation of these peptides. The association rate ($k_{on} = 1/\tau_{on} * C$) decreased with the voltage, indicating rapid translocation of peptides at higher

voltages (**Figure R10**). As suggested by the reviewer, we have included dwell time association rate histogram fitted with a monoexponential probability function (see main Figure 2). Also, we have included binding kinetics (k_{on} , k_{off} , and K_D) (see Table S1).

- ✓ Here, we defined the charge-selective molecule transport of cationic R9 through the pore. Based on the kinetics of R9 binding, we explained the physical properties of the pore, mainly the charge pattern of the pore. Notably, anionic D9 peptides did not block the pore confirming the role of negatively charged residues in controlling translocation. In the revised manuscript, we have clarified this (see main text page 9, 10, Figure 2 and Figure S7). To support the single-channel data, we have performed steered MD simulations to show the molecular transport of R9 peptides through DpPorA to highlight the charge-selectivity of the pore (main text page 18, 19 and Figure 6).

Figure R10: DpPorA for single-molecule sensing of cationic peptides
Interaction of nonaarginine (R9) with single DpPorA (10 μ M, cis) at -25 mV and -50 mV. The corresponding τ_{off} and τ_{on} dwell time histogram of R9 blocking fitted with a monoexponential probability function is shown.

4. In Fig. 4, a, b, authors show and analyze a single Gaussian peak, despite the clear-cut view of multiple transitions and spikes on the original traces.

We thank the reviewer for bringing this to our attention. The reviewer is correct about the clear-cut multiple transitions and spikes. The given unitary conductance histogram has been plotted based on 50 independent single-channel events and the mean unitary conductance was calculated to be ~ 0.95 nS at +200 mV. We added a representative current trace of a single DcWza insertion at +200 mV to show ~ 0.95 nS conductance. In the revised manuscript, we added the current amplitude histogram for single DcWza insertion and ejection to explain the multiple transition spikes and modified figure 5 (**Figure**

R11). Notably, we observed multiple transitions and spikes, specifically at high voltages, most likely due to the gating. It is important to note that the DcWza peptide inserted into the planar lipid bilayer and formed pores only at high applied voltages. Specifically, DcWza peptides formed transient pores in the lipid membrane. In the revised manuscript, we discussed and clarified this (main text page 15, 16, Figure 5, Figure S10 and Figure S11).

Figure R11: Electrical recordings of DcWza. Single DcWza insertion showing multiple spikes at +200 mV and corresponding current amplitude histogram. Ejection of DcWza pore from the membrane showing multiple spikes at +200 mV and corresponding current amplitude histogram.

5. On line 394, authors state that ‘..The minimal average radii are 4.97 and 4.02 Å for LpPorA and DpPorA..’. Yet, from the analysis of data shown in Fig. 2, c, authors conclude ‘..an increasing voltage confirmed the translocation of these cationic peptides through the pore..’. Considering that the peptide diameter is most likely larger than nanopores diameter, how can the assertion stand correct? A more in-depth analysis would be required. Also, I recommend authors to present the voltage-dependent analysis in detail.

✓ We thank Reviewer 2 for this suggestion. The minimal average radii for the LpPorA and DpPorA was calculated to be 4.97 and 4.02 Å using HOLE. Consistent with this, the LpPorA and DpPorA exhibited different single-channel conductance of ~4.0 nS and ~1.5 nS, respectively (see Figure S4 and Figure 7). Next, we examined the interaction of R9 with DpPorA. Applied voltage electrostatically drives these cationic peptides into the pore and facilitates their electrostatic binding with the negatively charged residues in the pore. Specifically, we observed that the dissociation rate constant ($k_{\text{off}} = 1/\tau_{\text{off}}$) increased with increasing voltages, indicating dissociation by translocation of peptides. Based on this, we conclude that DpPorA formed a large pore facilitating the charge-selective voltage-dependent translocation of bulky R9 peptides. We quantified the voltage-dependent translocation kinetics (k_{on} , k_{off} , and K_D) and included this in the revised manuscript as suggested by the reviewer (main text page 9, 10, main figure 2, Figure S7 and Table S1).

✓ Additionally, we investigated the interaction of a smaller cationic peptide, tetraarginine R4, with the DpPorA, which is observed to translocate rapidly. In 1 M KCl, R4 did not produce any ion current blockages due to weak binding with the pore. Next, we investigated the interaction of R4 with the pore at a low salt electrolyte buffer (0.15 M KCl). Here, R4 binds to the pore and produces short blockage events confirming substantial electrostatic contribution due to charge screening (**Figure R12**). This data is consistent with the large size of the DpPorA pore. We have added this new data in the revised manuscript (main text page 10 and Figure S7).

Figure R12: *DpPorA for charge-selective sensing of cationic peptides*
a) Interaction of nonaarginine (R9) with single DpPorA (10 μ M, cis) at -50 mV. b) Interaction of nonaaspartate (D9) with single DpPorA (10 μ M, cis) at +50 mV. c) Interaction of tetraarginine (R4) with single DpPorA (10 μ M, cis) at -50 mV. d) Interaction of R4 with single DpPorA (10 μ M, cis) at -50 mV in the low salt buffer.

Minor points: 6. No description on how the nanopores selectivity was measured.

We thank the reviewer for the suggestion. We have included a detailed description in the revised manuscript.

“The ion selectivity measurements were performed by establishing a KCl concentration gradient across the bilayer chamber (1M KCl, cis and 0.15M KCl, trans). The potential difference was applied through Ag/AgCl electrodes with agarose salt bridges. The pore formation in the membrane resulted in the current at 0 mV. Subsequently, this current was manually set to zero by adjusting the applied voltage. The voltage required to achieve zero current was referred to as the 'reverse potential' (V_m), which can be used to calculate the permeability ratio of K^+ and Cl^- ions across the pore. The ion selectivity of the pores has been characterized by the Goldman-Hodgkin-Katz equation.

$$V_m = \frac{RT}{F} \ln \left(\frac{P_{K^+} [K^+]^{cis} + P_{Cl^-} [Cl^-]^{trans}}{P_{K^+} [K^+]^{trans} + P_{Cl^-} [Cl^-]^{cis}} \right)$$

In this equation, R is the universal gas constant ($8.314 \text{ J.K}^{-1}.\text{mol}^{-1}$), T is the temperature in Kelvin, F is the Faraday's constant (96485 C.mol^{-1}), P_K is the membrane permeability for K^+ , P_{Cl} is the relative membrane permeability for Cl^- , $[K^+]^{cis}$ is the concentration of K^+ in the cis side, $[K^+]^{trans}$ is the concentration of K^+ in the trans side, $[Cl^-]^{cis}$ is the concentration of Cl^- in the cis side and $[Cl^-]^{trans}$ was the concentration of Cl^- in the trans side. The reverse potential for DpPorA was calculated to be $+32 \text{ mV}$ (main text page 6, 24 and SI text).

7. There is plenty of previous work regarding the particular aspects of peptide-nanopore interactions; it would be very helpful for the interested readers a more comprehensive citation of such work and concise analysis and discussion of authors data vs. what's already established in this field, that could indeed lead to '..new functional nanopore biosensors..'.
 ✓ We appreciate the reviewer for this suggestion which significantly improved the manuscript. Transmembrane pores based on α -helices have remained relatively unexplored and are emerging as a hot topic in nanopore technology. (Mravic et al. Science 2019, Xu et al. Nature 2020 and Scott et al. Nature Chemistry 2021). Applications in nanobiotechnology are emerging for new nanopores in sensing and sequencing. Here, we show that simple peptide chemical synthesis allowed us to build structurally stable pores of desirable charge patterns that readily inserted into lipid bilayers. Such autonomous α -helix-based assemblies would add new structural motifs to the field and pave the way to new nanopore sensors. We emphasize that high charge selectivity can be achieved by fine-tuning the helical pattern due to numerous side-chain interactions dominating the helix (main text page 2-4, 22-23).

✓ We defined the charge-selective translocation of polypeptides, cyclic sugars and large polymers across alpha-helical pores. We emphasize that this new class of alpha-helical pores is one of the rare synthetic channels to show very large conductance and cation selectivity and expand the scope of sensing complex biomacromolecules (main text page 4-10, Figure 1, Figure 2, Figure S2-S7). To demonstrate the functional versatility of the pore, we have included new experimental data to show DpPorA as a nanoreactor for single-molecule covalent chemistry (main text page 10, Figure 2 and Figure S6).

✓ As suggested by the reviewer, we have included a more comprehensive citation of previous work (peptide-nanopore interactions) and compared it with our work to develop a new functional nanopore sensor (main text page 22, 23 and references).

1. Hu, Z.L., Huo, M.Z., Ying, Y.L. & Long, Y.T. Biological Nanopore Approach for Single-Molecule Protein Sequencing. *Angewandte Chemie* 60, 14738-14749 (2021).

2. Asandei, A. et al. Electroosmotic Trap Against the Electrophoretic Force Near a Protein Nanopore Reveals Peptide Dynamics During Capture and Translocation. *ACS applied materials & interfaces* 8, 13166-13179 (2016).

3. Larimi, M.G., Mayse, L.A. & Movileanu, L. Interactions of a Polypeptide with a Protein Nanopore Under Crowding Conditions. *ACS nano* 13, 4469-4477 (2019).

4. Asandei, A. et al. Nanopore-based protein sequencing using biopores: current achievements and open challenges. *Small Methods* 4, 1900595 (2020)

Reviewer 3 comments

This manuscript is a breath of fresh air and I have read it with great interest. The Authors introduce the importance of stereochemistry to the structure and function of nanopores, which I consider a very important contribution. Peptide synthesis was used to create pores using D-amino acids that were characterized along with nanopores composed of L-amino acids. The Authors designed a set of clever experiments that demonstrated conductance and selectivity of both types of nanopores as well as application of the pores in resistive-pulse experiments. The measurements together with the MD simulations allowed them to unravel distribution of charged amino acids along the pore length and how this distribution facilitates detection of cationic species. Clear differences in the conductance of L and D pores were revealed that were explained by MD modeling. Finally, a striking difference was demonstrated in the stability of these nanopores to protease reactions. This manuscript certainly belongs to *Nature Comm.* and is bound to be cited a lot. I am convinced the nanopore structures the Authors presented will be used in fundamental studies as well as biotechnology. It is a very comprehensive work. I have only one minor comment – caption to Figure 1 does not provide all information; especially it is not clear which concentration gradient was used to obtain data in panel (d).

We want to thank the reviewer for recognizing the importance of this study and appreciating our work. We have added a detailed methodology of selectivity measurements in the revised manuscript (see main text page 6, 23 and SI text).

REVIEWER COMMENTS

Reviewer #1 (Remarks to the Author):

The authors demonstrated the functionalization of two types of pore-forming peptides and their chiral inverse counterparts. Compared to the previous submission, it is convincing that the chiral pore-forming materials are functionalized as a pore.

However, still a lack of novelty in fundamental sensing principles and mechanisms as a single molecule detection tool considering the following reasons. The author oversells their results on sensing. The manuscript did not novelty enough to meet the high quality of Nature Communications. Therefore, this manuscript is more suitable for a more specified journal, such as biophysics journals:

1. PorACj and DpPorA:

a) A higher affinity is observed with am δ yCD, and has been attributed to stereo inversion. This observation alone is very remarkable, but the relation between a stereo distribution and physical chemistry behind affinity is missing.

b) A thiol-containing PEG blocker is tested with the DpPorA and the disulfate bond is demonstrated to be broken. Do this phenomenon reproducible with PorACj? If I would have been the authors, it more preferable to study the D-Cys in both pores to investigate the influence of the stereo distribution?

c) DpPorA is resistant to proteinase K, this might be very nice merit of such D-type pore-forming materials for applications such as in vivo detection or drug delivery in a harsh environment. But the authors just stated that such pore-forming material is resistant to proteinase K, as quite obvious that ubiquitous enzyme is more conformation specified rather than insist with primary sequence information.

2. LcWza and DcWza

a) From the results, the authors demonstrated that LcWza is a less noisy pore, but unstable to proteinase K. In contrast, DcWza is noisier, and resistant to proteinase K. The authors seem not able to overcome this see-saw problem between stable pore and quite a pore. The results are quite preliminary.

b) One of the outcomes stated: Our data establish the stability of D-peptide pores to protease degradation irrespective of their amino acid sequence. As a chiral pore-forming material, it is okay, but such a statement is definitively not sufficient as a novel sensing tool.

3. The above-mentioned two types of pore-forming materials are quite segregated.

4. The writing style is not suitable for a broad readership, such as: Nevertheless, there is tremendous interest in building α -helical transmembrane pores since they offer a wide range of functional properties that cannot be achieved with redesigned β -barrel pores²⁹⁻³². There should specify what are the advantages.

I am convinced that the authors showed the reproducibility of the pore-forming activity from D-peptide. But the stitching skill to merge the two pieces of work is superficial. More important, there lacks guidance or concepts on how to de novo design a biosensor using the D-form materials.

Reviewer #3 (Remarks to the Author):

The Authors addressed all critical comments of all reviewers, and I am very happy to recommend this manuscript for publication. I do think this is a very important contribution to biomimetic nanopores and nanopore analytics.

We would like to thank all reviewers for their thoughtful comments. We have addressed each comment and made substantial changes that have significantly improved the quality of the revised manuscript.

Response to Reviewer 1

The authors demonstrated the functionalization of two types of pore-forming peptides and their chiral inverse counterparts. Compared to the previous submission, it is convincing that the chiral pore-forming materials are functionalized as a pore.

However, still a lack of novelty in fundamental sensing principles and mechanisms as a single molecule detection tool considering the following reasons. The author oversells their results on sensing. The manuscript did not novelty enough to meet the high quality of Nature Communications. Therefore, this manuscript is more suitable for a more specified journal, such as biophysics journals.

- ✓ We are happy to hear that the Reviewer recognizes our work demonstrating the pore-forming activity of the D-peptide and thank the Reviewer for their comment.
- ✓ We would like to highlight that the novel aspect of this work is the development of a well-defined mirror image peptide pore capable of charge selective sensing of cyclic sugars and peptides that have the stereochemical advantage of resistance to proteolytic degradation. Peptides built from natural amino acids are highly susceptible to proteolysis, limiting their applications. We emphasize that we have conducted a multidisciplinary approach to illustrate the D pore formation and outline its interaction with several analytes. We characterize the developed pores extensively using single-channel recordings. This manuscript is the first successful report of functional transmembrane pores built entirely from synthetic D-amino acid α -helical peptides and we conducted this work to pave the way for the development of sophisticated alpha-helical pores to provide new structural motifs for nanopore sensors.
- ✓ We would like to highlight that transmembrane alpha-helical pores remain relatively unexplored and have recently emerged as a hot topic in nanobiotechnology (Mravic et al. Science 2019, Xu et al. Nature 2020 and Scott et al. Nature Chemistry 2021). These papers are excellent pieces of work that focus on the structural aspects of alpha-helical pores outside a membrane environment. However, studies on single-channel recordings to illustrate pore characteristics are currently limited.

- ✓ We have in this study illustrated the interaction of the following analytes to show the versatility of the designed pore and our electrophysiology experiments are complemented by MD simulations.
 - Cationic γ -cyclodextrin ($\text{am}_8\gamma\text{CD}$)
 - Neutral γ -cyclodextrin (γCD)
 - Anionic γ -cyclodextrin ($\text{s}_8\gamma\text{CD}$)
 - Nona-arginine (R9)
 - Tetra-arginine (R4)
 - Nona-aspartate (D9)

- ✓ We would also like to highlight that compared to biological nanopores such as alpha-hemolysin, the reported mirror pores may be fabricated using solid-phase synthesis and can therefore be readily modified to incorporate amino acids of interest. Additionally, the purification process is simplified in this system due to the spontaneous self-assembly of DpPorA oligomers. The synthetic alpha-helical pores described in this study also undergo easier insertion into the membrane than DNA pores. Such versatility, ease of synthesis, and ready membrane insertion are highly desirable properties of a nanopore for investigating single-molecule chemistry and applications in biotechnology. We believe this work is highly original system of outstanding general interest due to the unprecedented pore architecture, the significance of building better nanopores, and their potential applications for nanopore technologies. Therefore, we strongly believe that our findings in the revised manuscript are eminently suited to Nature Communications.

1. PorACj and DpPorA:

1a) A higher affinity is observed with $\text{am}_8\gamma\text{CD}$, and has been attributed to stereo inversion. This observation alone is very remarkable, but the relation between a stereo distribution and physical chemistry behind affinity is missing.

We thank the Reviewer for raising this important question. Stereo-inversion appears to have resulted in a constriction of the pore and differences in the electrostatic distribution, as indicated by MD simulations in Figure 7 in the main manuscript. The higher affinity of $\text{am}_8\gamma\text{CD}$ with DpPorA as compared with LpPorA is attributed to enhanced interaction owing to the narrowing of the corresponding pore. To better illustrate the molecular basis behind the differences in interaction, we have included new data showing steered MD simulations of $\text{am}_8\gamma\text{CD}$ with LpPorA (PorACj).

Interestingly, unbiased MD simulations indicated that the binding of cationic $\text{am}_8\gamma\text{CD}$ with pPorA was stabilized by strong hydrogen bonding and electrostatic interactions with the carboxyl groups of the aspartic acid residues. The permeation of the cationic $\text{am}_8\gamma\text{CD}$ through LpPorA and DpPorA was explored using steered MD (SMD) simulations to estimate the force required to enable the transport process. The SMD force profiles were generated as the function of the reaction coordinate z , i.e., the center of mass (COM) distance between $\text{am}_8\gamma\text{CD}$ and the C_α atoms of the pPorA, by pulling the ligand molecule along the channel axis of pPorA. The average maximum force required to pull the cationic $\text{am}_8\gamma\text{CD}$ through DpPorA was ~ 400 pN and LpPorA was ~ 370 pN. The relatively larger force required to pull the $\text{am}_8\gamma\text{CD}$ through DpPorA compared to that of the LpPorA corroborate the greater binding affinity of $\text{am}_8\gamma\text{CD}$ with DpPorA, likely due to the slightly larger LpPorA pore diameter. We have discussed this data in the revised manuscript, which is in line with experimental data (main text page 18-20, figure 6, 7 and new figure S15).

Figure R1. Permeation of $\text{am}_8\gamma\text{CD}$ through LpPorA and DpPorA
a) Transport of $\text{am}_8\gamma\text{CD}$ through LpPorA. **b)** Average force profile from steered MD simulation for the permeation of $\text{am}_8\gamma\text{CD}$ as a function of reaction coordinate z , i.e., the center of the mass distance between $\text{am}_8\gamma\text{CD}$ and the C_α atoms of LpPorA. **c)** Transport of $\text{am}_8\gamma\text{CD}$ through DpPorA. **d)** Average force profile from steered MD simulation for the permeation of $\text{am}_8\gamma\text{CD}$ as a function of reaction coordinate z , i.e., the center of the mass distance between $\text{am}_8\gamma\text{CD}$ and the C_α atoms of DpPorA.

1b) A thiol-containing PEG blocker is tested with the DpPorA and the disulfate bond is demonstrated to be broken. Do this phenomenon reproducible with PorACj? If I would have been the authors, it more preferable to study the D-Cys in both pores to investigate the influence of the stereo distribution?

We have indeed conducted similar experiments with LpPorA and would like to thank the Reviewer for the opportunity to discuss these results. We observed reproducible stepwise pore blockage in LpPorA (PorACj) on the addition of the thiol-containing blocker PEG-OPSS 1K. This has now been discussed and we have cited the relevant reference in the revised manuscript (see main text page 10).

Figure R2. Site-specific chemical modification of pPorA pores with PEG polymers

a) Electrical recordings showing the reversible chemical modification of 1 mM MePEG-OPSS-1k with DpPorA at +50 mV. **b)** Electrical recordings showing the reversible chemical modification of 1 mM MePEG-OPSS-1k with LpPorA at +50 mV. The current signals were digitally filtered at 500 Hz.

To build further on this aspect, we studied the interaction of the larger thiol blocker PEG-OPSS 5k with DpPorA and compared it to that with LpPorA. LpPorA has previously been shown to interact with PEG-5k to undergo complete single-step pore closure. In contrast, no blockage was observed with DpPorA. This result further highlights the constricted pore size of DpPorA compared to LpPorA. Owing to the significantly larger hydrodynamic radius of PEG-OPSS-5k, the molecule cannot penetrate the smaller DpPorA. However, in the larger LpPorA, the cysteine residues remain accessible for large PEG-5k. This new data has been included in the revised manuscript. We have also cited the relevant reference (main text page 8-10 and new figure 2). The recommendation to specifically modify the cysteine residue in the sequence is an interesting suggestion and we will be working on this in the future.

a) DpPorA with MePEG-OPSS-5K (MW: 5000 Da) at +50mV b) LpPorA with MePEG-OPSS-5K (MW: 5000 Da) at +50mV

Figure R3. Site-specific chemical modification of pPorA pores with large PEG polymers
a) Electrical recordings showing no chemical modification of 1 mM MePEG-OPSS-5k with DpPorA at +50 mV. **b)** Electrical recordings showing the reversible chemical modification of 1 mM MePEG-OPSS-5k with LpPorA at +50 mV. The current signals were digitally filtered at 500 Hz.

1c) DpPorA is resistant to proteinase K, this might be a very nice merit of such D-type pore-forming materials for applications such as in vivo detection or drug delivery in a harsh environment. But the authors just stated that such pore-forming material is resistant to proteinase K, as quite obvious that ubiquitous enzyme is more conformation specified rather than insist with primary sequence information.

- We sincerely thank the reviewer for highlighting the importance of such D-type pores as described in this manuscript for applications in in vivo detection in harsh environments. This is a particularly important statement and has now been included in the introduction and discussion section of the revised manuscript (see main text page 4 and 23).
- We would also like to highlight that DpPorA remains functional after Proteinase K treatment. We show that the protease treated DpPorA exists in a stable structural conformation facilitating the selective binding of cationic $\alpha\text{-MCD}$ with applied voltage serving as a driving force. To highlight this, we have included new data showing CD blocking at different voltages. We emphasize that these pores, as charge-selective single-molecule sensors possess the stereochemical advantage of resistance to proteolytic degradation (main text page 11, 12 and new figure 3).

Figure R4. Single-molecule sensing of protease treated DpPorA
Interaction of proteinase K reacted DpPorA with am₈γCD (10 μM, trans) at a) +25 mV, b) +50 mV and c) +75 mV showing functional stability of the pores. All current signals were filtered at 2 kHz and sampled at 10 kHz.

- In addition to the single-channel recording data already provided, we have added new data from fluorescence assays to visualize the incorporation of DpPorA treated with proteinase K in giant unilamellar vesicles (GUVs). We observed the transport of Alexa dye inside the vesicle lumen represented by fluorescence intensity $I_{in} \approx I_{out}$ (where the vesicles show fluorescence inside their lumen), as compared to control. The vesicle permeabilization rate was estimated to be $88 \pm 1.4\%$ (mean \pm SD from n= 100 vesicles), whereas that of the control was only $5 \pm 1.4\%$ (n = 100 vesicles). The results confirm stable pore conformation in an alternate membrane model (main text page 15 and new figure 4).

Figure R5. Functional assembly of DpPorA in giant vesicles
a) Fluorescence image of single vesicle in the absence of DpPorA and incubated with proteinase K treated DpPorA for 15 minutes displaying dye transport. Insets display the contour of vesicles in fluorescence. **b)** Vesicle permeabilization rates in the presence of proteinase K treated DpPorA and absence of DpPorA. Buffer conditions: 100 mM KCl, 10 mM HEPES pH 7; scale bar: 10 μm.

2. LcWza and DcWza

2a) From the results, the authors demonstrated that LcWza is a less noisy pore, but unstable to proteinase K. In contrast, DcWza is noisier, and resistant to proteinase K. The authors seem not able to overcome this see-saw problem between stable pore and quite a pore. The results are quite preliminary.

Reviewer 1 has raised an important point in this comment. We would like to highlight that the purpose of this study is to present DpPorA as a mirror peptide pore that is resistant to Proteinase K and also as functional and stable as its L-counterpart. The purpose of including L and DcWza in the current work is to illustrate that this is not a ubiquitous property or a common feature for all pore-forming peptides. Therefore, the motivation behind the study of DcWza was to demonstrate the importance of sequence specificity in the ability of D peptides to form stable pores in lipid bilayers. We have highlighted this by showing several contrasting pore-forming properties between DpPorA and DcWza. More specifically, DpPorA pore formation occurs by insertion of self-assembled octamers, whereas DcWza forms pores in the lipid membrane by the membrane-induced association of peptide monomers. Despite the differences in pore formation, both DpPorA and DcWza are shown to be stable to proteases, demonstrating the advantages of stereo-inversion to enhance the proteolytic resistance of pore-forming peptides. However, this does not indicate that the stereo-inversion of amino acids will result in a stable pore and therefore demonstrates sequence specificity. We have discussed and clarified this in the revised manuscript (main text page 2-4, 15-17 and 23).

The Reviewer also raised the concern that the cWza data was preliminary. Although the cWza system was provided to highlight the quality of the major system, we have included new data and improved data analysis in the revised manuscript to definitively elucidate cWza pore formation and avoid any ambiguity. We provided a plot based on the statistical analysis of 50 single DcWza insertion events to show the presence of different DcWza populations, including stable pores. Additionally, we have included stable current traces of DcWza pores at +50 mV and +100 mV and discussed this in the revised manuscript (new figure 5, figure S10, S11, S12, main text page 15-17).

Notably, the observations in single-channel recordings for both sets of pores closely correspond to the MD simulation data.

Figure R6. Single-channel properties of DcWza.

a) Single DcWza in a stable conductance state at +50 mV and **b)** at +100 mV. **c)** plot representing different DcWza populations (stable in blue, unstable in red and closed states in green). The current signals were filtered at 2 kHz and sampled at 10 kHz.

2b) One of the outcomes stated: Our data establish the stability of D-peptide pores to protease degradation irrespective of their amino acid sequence. As a chiral pore-forming material, it is okay, but such a statement is definitively not sufficient as a novel sensing tool.

The motivation behind the study of cWza was to demonstrate

- ✓ the importance of sequence specificity in the ability of D peptides to form stable pores in lipid bilayers by demonstrating several contrasting pore-forming properties between DpPorA and DcWza
- ✓ to demonstrate the advantages of stereo inversion to enhance proteolytic resistance as evidenced by their stability to protease.

We have clarified this in the revised manuscript (main text page 2, 15-17 and 23).

We would like to highlight that the novel aspect of this work is the development of a well-defined charge-selective mirror image DpPorA pore that has the critical stereochemical advantage of resistance to proteolytic degradation used for single-molecule sensing of cationic $\alpha\text{-m}_3\gamma\text{CD}$. GUV fluorescence assays complement our single-channel recordings. In the revised manuscript, we have clarified this (see main text pages 4, 12, 15-17, 23 and new figures 3 and 4).

3. The above-mentioned two types of pore-forming materials are quite segregated.

As highlighted in response to comment 2(a), the motivation behind the study of cWza was to demonstrate the importance of sequence specificity in the ability of D peptides to form stable pores in lipid bilayers by studying the stereo-inversed counterparts of two pore-forming peptides.

We believe that this data is essential in this study to highlight these aspects. However, based on the recommendation of Reviewer 1 that the two aspects of this study appear segregated, we have revised the text to clearly outline the purpose, improve the presentation of the study as well as avoid ambiguity in the manuscript (see main text page 2-6, 12, 15-17, 23 and new figure 3, new figure 4, new figure 5, figure S10, S11 and S12).

4. The writing style is not suitable for a broad readership, such as: Nevertheless, there is tremendous interest in building α -helical transmembrane pores since they offer a wide range of functional properties that cannot be achieved with redesigned β -barrel pores²⁹⁻³². There should specify what are the advantages.

We thank the Reviewer for bringing this crucial point to our attention. We are committed to ensuring our manuscript is accessible to a broad readership and have carefully gone through the entire manuscript to modify such instances in the text.

The quoted sentence was modified as follows: “Nevertheless, there is tremendous interest in building α -helical transmembrane pores since they offer a wide range of functional properties similar to natural membrane proteins²⁹⁻³². Importantly, discovering alpha-helical pores with defined structures that selectively conduct ions and molecules could lead to the development of new functional nanopore biosensors^{1, 33} “(see main text page 3).

5. I am convinced that the authors showed the reproducibility of the pore-forming activity from D-peptide. But the stitching skill to merge the two pieces of work is superficial. More important, there lacks guidance or concepts on how to de novo design a biosensor using the D-form materials.

Thank you. We emphasize that our findings will aid the design of a new class of alpha-helical pores for applications, including sensing and sequencing. **Also, please see our detailed response to the first comment regarding the novelty of the work.**

Response to Reviewer 3:

The Authors addressed all critical comments of all reviewers, and I am very happy to recommend this manuscript for publication. I do think this is a very important contribution to biomimetic nanopores and nanopore analytics.

Many thanks for recognizing the importance of the study.

REVIEWER COMMENTS

Reviewer #1 (Remarks to the Author):

The manuscript entitled “Assembly of Transmembrane Pores from Mirror-Image Peptides” by Mahendran et al. proposed the mirror-image peptide formed transmembrane pores. The authors respond the previous suggestions with more results showing the formation of transmembrane pores composed from mirror-image peptides. As already mentioned in the previous comments that I totally agree with the pore formation, however, the general scope of the manuscript is still too narrow for broad journal as Nature communications for the following reasons:

1. The authors emphasize the importance of sequence specificity in the ability of D peptides to form stable pores in lipid bilayers. After careful examine the two pores forming peptide whose sequences are, DpPorA: midqiteifgqlgtflggfngnifcglkdvietivkwaak; DcWza: aplvrwnrvisqlvptitgvhdltetvrciktwpn, I did not find the fidelity of such so-called specific sequence.
2. The authors summarize the two segregate stories on DpPorA and DcWza towards the specific sequence for stable pore forming. As mentioned in the first comments, there is still no link of such sequence, the authors lack the ability to concise the scaffold of their manuscript.
3. In previous response, the authors were questioned about the reproducibility of disulfate bond forming in DpPorA and PorACj, in addition, D-Cys interaction with both pores. The authors put more results on the fore-part of the comments but failed to show the rare-part. It seems synthesis of a whole D-amino acid peptide is applicable, but the single D-Cys mutant is not aware in such critical point?
4. In previous comments, “2a) From the results, the authors demonstrated that LcWza is a less noisy pore, but unstable to proteinase K. In contrast, DcWza is noisier, and resistant to proteinase K. The authors seem not able to overcome this see-saw problem between stable pore and quite a pore. The results are quite premilitary.” The authors failed to answer this question and trying to redirect the response to their “sequence specificity” key points.

In the end, the authors trying to show the D peptide can form a transmembrane pore which is not novel. As the authors mentioned 3 nice paper to emphasize the importance of transmembrane alpha-helical pores, Mravic et al. Science 2019, proposed the importance of polar and apolar sidechains in stabilizing the pore structure; Xu et al. Nature 2020, addressed computational design of two concentric rings of α -helices that are stable and monodisperse in both their water-soluble and their transmembrane forms; Scott et al. Nature Chemistry 2021 use rational de novo design to generate water-soluble α -helical barrels with polar interiors, then they modify the sequence for better membrane insertion. All these nice paper showed a comprehensive de novo design for a stable transmembrane alpha-helical structures. The work in this manuscript is crude, neither proper design of the sequence to enhance the structure, nor to make a solid compare between L-peptide and D-peptide, whose sensing capability is better and the biophysics running behind. So I strongly recommend the manuscript is more suitable for more specific targets and showed a nice results on chiral material for transmembrane pore formation.

We want to thank the Reviewer for their comments and suggestions. We have taken the Reviewer's comments seriously and addressed all concerns in the current round of revision.

Response to Reviewer 1

The manuscript entitled "Assembly of Transmembrane Pores from Mirror-Image Peptides" by Mahendran et al. proposed the mirror-image peptide formed transmembrane pores. The authors respond the previous suggestions with more results showing the formation of transmembrane pores composed from mirror-image peptides. As already mentioned in the previous comments that I totally agree with the pore formation, however, the general scope of the manuscript is still too narrow for broad journal as Nature communications for the following reasons:

Thank you. In the previous round of revision, we have addressed the Reviewer's initial concerns in detail, supported by additional experiments that we believe the reviewer has now accepted.

Regarding the suitability of this work for Nature communications,

- ✓ We emphasize that this is the first stable, functional large transmembrane pore built entirely from synthetic D-amino acid α -helical peptides.
- ✓ We probed the structure, assembly and functional properties of these pores using a combination of peptide redesign and synthesis, solution-phase biophysics, single-channel recordings, fluorescence vesicle transport assays and molecular dynamics simulations.
- ✓ The pores demonstrated here are a highly original system of outstanding general interest due to the alpha-helical protein architecture and their potential applications in nanopore technology and nanopore chemistry. Please note that most previous efforts in this area have focused on β -barrel proteins, solid-state nanopores, and DNA pores.

We believe our paper, which is an interdisciplinary work with important implications in single-molecule biophysics, nanoscience, and chemical biology, will be of interest to the broad readership of Nature Communications.

1. The authors emphasize the importance of sequence specificity in the ability of D peptides to form stable pores in lipid bilayers. After careful examine the two pores forming peptide whose sequences are, DpPorA: midqiteifgqlgtflggfngnifcglkdvietivkwaak; DcWza:

aplvrwnrvisqlvptitgvhdltetvrciktwpn, I did not find the fidelity of such so-called specific sequence.

- ✓ Our approach was to engineer transmembrane alpha-helical pores based on natural alpha-helical assemblies and build the corresponding functional peptide pores using chemical synthesis, which is highlighted in the manuscript. Specifically, we constructed peptides based on the natural pores Wza and PorACj. The two peptide sequences are distinct, and show significant differences in hydrophobicity, where pPorA, (based on PorACj) is predominantly hydrophobic and cWza (based on Wza) is amphipathic (main text pages 3 and 4).
- ✓ Notably, the K24C mutant of pPorA autonomously assembles into an SDS stable pre-oligomer which inserts into the membrane to form large transmembrane pores of uniform conductance. Again, we would like to highlight that this is one of the rare channels that undergoes such pre-oligomerization to form SDS-stable assemblies. In contrast, cWza has a completely different pore assembly mechanism wherein the monomeric peptides undergo membrane-associated pore formation (main text pages 6 and 23).
- ✓ In our study, we investigated the behavior of the stereo-inversed versions of these two peptides. Single-channel electrophysiology showed that the extracted octamer of DpPorA inserted into the membrane to form stable and functional pores. The pores show characteristic blocking upon interaction with cationic peptides and cyclic sugars, which can find applications in nanopore technology. In contrast, DcWza only formed transient pores upon insertion in the lipid membrane. The pores were not functional, and no blockages were observed upon adding cationic $\text{am}_8\gamma\text{CD}$. This trace is now included in the revised supporting information to support this statement (main text page 17 and Figure S11). Our results clearly show the distinct behavior of two chemically synthesized peptides, highlighting the importance of the peptide sequence in stable, functional pore formation.

Figure R1: Electrical and functional properties of transient DcWza pores
Electrical recording of transient DcWza that remained in the open conductance state in the absence and presence of 100 μM (trans) $\text{am}_8\gamma\text{CD}$ at +100 mV. The current signals were filtered at 2 kHz and sampled at 10 kHz. Electrolyte: 1 M KCl, 10 mM HEPES, pH 7.4.

- ✓ We also conducted an additional experiment to assess the pore formation ability of DcWza in peptides in giant unilamellar vesicles (GUVs). DDM solubilized DcWza peptides were added to pre-formed GUVs and the uptake of the hydrophilic fluorescent dye Alexa-Fluor 350 across the vesicles was then monitored over time. We observed that the fluorescence intensity remained constant over time, providing further evidence that DcWza does not form stable pores and does not permit the passage of dye molecules. This data has been included in the supporting information (main text pages 16, 17 and Figure S12).

Figure R2: Transport across DcWza pores in giant vesicle system

Fluorescence image of a single vesicle revealing dye transport at 60 minutes in the presence and absence of DcWza (control), where false blue color represents the Alexa-Fluor 350 dye in vesicles, inset shows the image of the vesicle (fluorescently labeled with 0.05 mol% ATTO-550 DOPE). Scale bar: 10 μ m. Comparison of vesicle permeabilization rates in the absence and presence of peptide DcWza. Buffer conditions: 100 mM KCl, 10 mM HEPES pH 7.

- ✓ Please note that β -barrels have been engineered extensively. In contrast, transmembrane alpha-helical pores are relatively underexplored due to many challenges, including the ability of helices to rupture membranes non-specifically, which is partly due to the hydrophobic effect and the peptide sequence. Accordingly, this study demonstrates the sequence specificity between peptides (DpPorA and DcWza) in the stable pore-formation process. We have clarified this in the revised manuscript (main text pages 3,6,16, 17 and 23).

2. The authors summarize the two segregate stories on DpPorA and DcWza towards the specific sequence for stable pore forming. As mentioned in the first comments, there is still no link of such sequence, the authors lack the ability to concise the scaffold of their manuscript. As stated in response to the previous question, the highlight of this work, as recognized by Reviewers 2 and 3, was to introduce new alpha-helical motifs to build a better nanopore sensor. We have introduced DpPorA as an example of a large alpha-helical pore with high conductance blocked by several analytes. In contrast, the DcWza pore is an example of a peptide built from D-amino acids that do not form stable pores in lipid membranes, different from its L-counterpart.

There are several obstacles in the rational redesign of functional transmembrane helical bundles, including balancing the membrane solubility and self-assembly to give well-defined structures. We have presented two examples of peptide sequences based on natural porins that show highly contrasting pore formation behavior in the membrane environment. This highlights the challenges in designing alpha-helical peptides to obtain stable pores and shows the importance of the amino acid sequence in balancing hydrophobic and multiple side-chain interactions to stabilize their folding and assembly. We have discussed this in the revised manuscript (main text pages 3,6,16, 17, 23, Figure S11 and Figure S12).

3. In previous response, the authors were questioned about the reproducibility of disulfate bond forming in DpPorA and PorACj, in addition, D-Cys interaction with both pores. The authors put more results on the fore-part of the comments but failed to show the rare-part. It seems synthesis of a whole D-amino acid peptide is applicable, but the single D-Cys mutant is not aware in such critical point?

- ✓ For all new pores, it is essential to show ion flow through the expected pathway. In one approach, thiol groups are placed in the potential pathway as sites for blockers. Activated thiols react with these sites to form disulfide bonds. The advantage of this system is that disulfide bonds can be specifically cleaved in the presence of DTT, which is highly advantageous in ruling out non-specific loss of activity. In the current system, DpPorA contains cysteine residues at the 24th position, which is expected to be in the lumen of the pore. Activated PEG thiols (MePEG-OPSS-1k) reacted with the thiol groups of the cysteine residue leading to a blockage in the current flow. As stated previously, blockages occurring due to disulfide bond formation are reversed by adding DTT. This data was included in the original submitted manuscript (main text pages 10, 11, Figure 2d and Figure S8).
- ✓ We have included two new experiments (+50 mV and +25 mV) in the revised supporting information showing the pore closure of DpPorA in the presence of MePEG-OPSS-1k and the subsequent pore reopening with DTT to establish the reproducibility of disulfide bond formation.

New experiment at +50 mV

In our chemical modification experiments, we first recorded the current traces of the pore in the absence of methyl-PEG-OPSS-1k at +50 mV, where DpPorA remains in the open state. When mPEG-OPSS-1k (1mM, cis) is added to the same pore, we observe that the pore closes in several steps, indicating a current drop due to the interaction of the PEG-OPSS with the -SH groups of the cysteine residues. Upon the addition of DTT (10 mM) to the same pore, the PEG polymers were cleaved, and the pore reverted to the open

conductance state. Importantly, the channel remains closed and reverts to an open state only after adding DTT. This clearly demonstrates that pore closure is due to a highly specific interaction involving the formation of a disulfide bond between mPEG-OPSS-1k and the side chain of the D-Cys in the 24th position. This also indicates that the SH side chains of the cysteine residues are present in the ion conductance pathway in the pore lumen (new figure S8b).

New experiment at +25 mV

An additional experiment showing mPEG-OPSS 1k induced pore closure and subsequent DTT-induced opening have also been shown at a lower voltage of +25 mV. DpPorA remained in the fully open conductance state at +25 mV. Further on, adding MePEG-OPSS 1k (1mM, cis) to the lipid bilayer chamber and spontaneously mixing the chamber contents resulted in the pore closure in several steps indicating chemical modification. Upon the addition of DTT (10 mM) to the same pore, the PEG polymers were cleaved, and the pore reverted to the open conductance state at +25 mV (new figure S8c).

- ✓ In order to further demonstrate that the pore was fully functional, we added 10 μ M cationic gamma CD ($\text{am}_8\gamma\text{CD}$) to the trans side of the chamber prior to the addition of mPEG-OPSS-1k. A distinct blockage pattern was observed, confirming pore functionality. The current trace showing analyte blocking has now been provided in Figure S8. The gamma CD was then perfused from the chamber and mPEG-OPSS was added to the cis side of the pore. After adding PEG-OPSS, we observed a stepwise current drop to a closed conductance state (new figure S8d).
- ✓ We thank the reviewer for recognizing the importance of designing and synthesizing the D-amino acid peptides. In our study, we targeted peptides built completely from D-amino acids and have successfully demonstrated a well-defined mirror image peptide pore capable of charge selective sensing of cyclic sugars and peptides that is also resistant to proteolytic degradation. We agree that studying the D-Cys mutant will be interesting for select applications. However, we believe that the complete mirror image peptide is more suitable for our goal of building motifs for better nanopore sensors.

a) Interaction of MePEG-OPSS with DpPorA

b) DpPorA with MePEG-OPSS-1K addition at +50 mV

DpPorA opening mediated by 10mM DTT at +50 mV

c) DpPorA with 1mM MePEG-OPSS-1K (cis) addition at +25 mV

DpPorA opening mediated by 10mM DTT at +25 mV

d) DpPorA with $\text{am}_8\gamma\text{CD}$ at +50 mV (trans)

Same DpPorA at +50 mV after perfusion of $\text{am}_8\gamma\text{CD}$

Same DpPorA with 1 mM MePEG-OPSS-1K (cis) at +50 mV

Figure R3: Interaction of DpPorA with PEG polymers

a-c) Schematic and electrical recordings (two independent experiments) showing the reversible chemical modification of 1 mM MePEG-OPSS-1k with DpPorA at +50 mV and +25 mV and the addition of 10 mM DTT resulted in the pore opening. **d)** Electrical recordings showing the interaction of $\text{am}_8\gamma\text{CD}$ with single DpPorA (10 μM , trans) at +50 mV and chemical modification of 1 mM MePEG-OPSS-1k with same DpPorA at +50 mV after perfusion of CDs. The current signals of the PEG experiments were digitally filtered at 500 Hz and $\text{am}_8\gamma\text{CD}$ interaction was filtered at 2 kHz.

4. In previous comments, “2a) From the results, the authors demonstrated that LcWza is a less noisy pore, but unstable to proteinase K. In contrast, DcWza is noisier, and resistant to

proteinase K. The authors seem not able to overcome this see-saw problem between stable pore and quite a pore. The results are quite preliminary.” The authors failed to answer this question and trying to redirect the response to their “sequence specificity” key points.

In the revised version of the manuscript, we have included the following new data (as mentioned in reply to the first comment) to further elucidate the contrasting behavior of DcWza and DpPorA.

- A recording of the addition of cationic am₈γCD (100 μM, trans) to the DcWza pore. No blockages were observed, showing that the pore did not interact effectively with the analyte (main text page 16,17 and Figure S11).
- Data showing dye transport across giant unilamellar vesicles in which DcWza was reconstituted. We observed that the fluorescence remained constant over time, providing further evidence that DcWza forms only transient pores and does not permit the passage of dye molecules (main text pages 16,17 and Figure S12).

While we observed that DcWza does not form a stable pore, we disagree with the reviewer that the results are preliminary. In our study, we have presented the mirror pore DpPorA as a stable and functional pore suitable for use as a nanopore sensor. As mentioned previously, DcWza has been presented as an example of a system that forms transient pores, unlike its L-counterpart. The data presented for DcWza pore formation is based on a statistical analysis of more than 100 experiments (main text 17 and Figure S12). We have presented the distinct behavior of the two peptides in single-channel recordings, SDS-PAGE, and dye permeation assays.

5. In the end, the authors trying to show the D peptide can form a transmembrane pore which is not novel. As the authors mentioned 3 nice paper to emphasize the importance of transmembrane alpha-helical pores, Mravic et al. Science 2019, proposed the importance of polar and apolar sidechains in stabilizing the pore structure; Xu et al. Nature 2020, addressed computational design of two concentric rings of α-helices that are stable and monodisperse in both their water-soluble and their transmembrane forms; Scott et al. Nature Chemistry 2021 use rational de novo design to generate water-soluble α-helical barrels with polar interiors, then they modify the sequence for better membrane insertion. All these nice papers showed a comprehensive de novo design for a stable transmembrane alpha-helical structures. The work in this manuscript is crude, neither proper design of the sequence to enhance the structure, nor to make a solid compare between L-peptide and D-peptide, whose sensing

capability is better and the biophysics running behind. So I strongly recommend the manuscript is more suitable for more specific targets and showed a nice results on chiral material for transmembrane pore formation.

- ✓ As the reviewer has recognized, the three excellent papers cited in the previous response discuss the de novo design of stable transmembrane alpha-helical structures. However, it is important to note that none of the designed channels formed pores of high ion conductance and charge selectivity in single-channel electrical recordings. Notably, none of these pores are made from D-amino acids.
- ✓ The first two studies (Mravic et al. Science 2019 and Xu et al. Nature 2020) focus on computational design and investigation of the structure of the designed alpha-helical assemblies. Although these papers are very impressive and represent the significant progress made in the computational design of peptide sequences that form transmembrane alpha-helical structures, single-channel electrical recordings that provide information about pore formation in the membrane environment have not been provided.
- ✓ The third study (Scott et al. Nature Chemistry 2021) reported the channel formation of the designed alpha-helical peptide in electrophysiology recordings; however, the pore mimics ion channels and has a very low conductance of ~ 0.12 nS, rendering it unsuitable for use as a nanopore sensor.
- ✓ Overall, we believe the development of alpha-helical pores with high conductance blocked by analytes and suitable for use as effective nanopore sensors is not well reported in the literature. We emphasize that DpPorA, designed based on the natural porin PorACj as an interesting chiral material for pore formation, as recognized by the reviewer, presents a new alpha-helical motif that will aid the design of such systems. Alpha helical nanopores sensors are not well explored and the stable and functional mirror image pore presented in this manuscript is the first of its kind. We have cited these papers and introduced our work (main text page 3 and 4).

Therefore, we firmly believe our study will be significant to researchers and scientists across a broad readership of Nature Communications.

REVIEWER COMMENTS

Reviewer #1 (Remarks to the Author):

The authors have revised the manuscript, provided more experimental evidence and addressed my questions. The manuscript is greatly improved, and is now more accessible to the reader. To further enhance the general interests of the manuscript, a few other points should be clarified.

-The importance of designing for alpha-helical pores

In Introduction, the authors claimed that "However, engineered alpha-helical pores for single-molecule sensing remain less explored. This is primarily because the engineering of alpha-helical pores often distorts their functional structure as multiple side-chain interactions within individual helices are required to stabilize their folding and assembly." Since the scope of this manuscript is to develop/design alpha-helical pores as nanopore sensor, the authors should also discuss the differences between beta-barrel pores and alpha-helical pores on single-molecule sensing.

-The importance of pore stability.

In the manuscript, the authors incorporated D-amino acids to natural sequence to increase the biostability of protein pores that can not be cleaved by protease enzymes, and stated it could "expand the scope of developing stable therapeutics". However, to my knowledge, the pore biostability is a double-edged sword for the applications in vivo, as the proteins that could not be digested means it would be more likely accumulated in the body. The authors should consider it and discuss more.

Response to Reviewer 1

1. The authors have revised the manuscript, provided more experimental evidence and addressed my questions. The manuscript is greatly improved, and is now more accessible to the reader. To further enhance the general interests of the manuscript, a few other points should be clarified.

We thank the Reviewer for their comment, especially on additional experiments. We have addressed all concerns in the current round of revision.

2. The importance of designing for alpha-helical pores

In Introduction, the authors claimed that “However, engineered alpha-helical pores for single-molecule sensing remain less explored. This is primarily because the engineering of alpha-helical pores often distorts their functional structure as multiple side-chain interactions within individual helices are required to stabilize their folding and assembly.” Since the scope of this manuscript is to develop/design alpha-helical pores as nanopore sensor, the authors should also discuss the differences between beta-barrel pores and alpha-helical pores on single-molecule sensing.

We would like to thank the Reviewer for this comment. Most single-molecule sensing studies have been done on the biological beta-barrel pores, such as alpha hemolysin and MspA. Several engineering manipulations have allowed alpha hemolysin to be developed as a powerful component for stochastic sensing (Ayub, M. & Bayley, H. 2016. *Curr. Opin. Chem. Biol.* 34, 117-126). However, these studies have led to only modest changes in the charge selectivity of ion transport (e.g., K^+ v Cl^-) through the alpha hemolysin.

However, since selectivity for cations or anions strongly depends on the pore surface charge pattern, modifications in α -helical pores can significantly alter the pore selectivity due to numerous side-chain interactions that occur within the helical structure. For example, the pPorA-K24 pore is anion-selective (P_{K^+}/P_{Cl^-} 1:3) and blocked by anionic analytes. Here, the mutation of the lysine at the 24th position to a cysteine residue results in a cation-selective pore (P_{K^+}/P_{Cl^-} 10: 1) blocked by cationic analytes (Krishnan, R.S. et al. 2019. *J. Am. Chem. Soc.* 141, 2949-2959). Thus, engineering channels to selectively detect specific ions or small molecules are vital for developing new biosensors. Also, helical bundles are exciting targets for single molecule sensing because of their similarities to several natural ion channels. Such pores might be used for sensing approaches that cannot be performed with beta-barrel pores, such as developing ion-selective channels. As suggested, we have discussed this in the

manuscript to highlight the differences between beta-barrel and alpha-helical pores for single molecule sensing (see main text page 3).

3. The importance of pore stability.

In the manuscript, the authors incorporated D-amino acids to natural sequence to increase the biostability of protein pores that cannot be cleaved by protease enzymes, and stated it could “expand the scope of developing stable therapeutics”. However, to my knowledge, the pore biostability is a double-edged sword for the applications *in vivo*, as the proteins that could not be digested means it would be more likely accumulated in the body. The authors should consider it and discuss more.

Our take-home message is that this paper is the first successful report of stable, functional membrane-spanning unnatural pores composed of D-amino acids.

We completely agree with the concern raised by the reviewer regarding the pore stability associated with developing therapeutics. The pore biostability in *in vivo* conditions can be challenging as the proteins would most likely accumulate in the body. However, we propose that these protease-stable D peptides can possess fewer immunological side effects. Based on the stability of designed D peptide pores in the presence of a protease, we suggest that such pores might be effective as antimicrobial and anticancer agents and will find applications in medicine. The *in vivo* activity of these peptides is yet to be explored and can be very challenging. Our results here provide a preliminary outlook on the possibility of exploiting these peptide pores as a therapeutic.

The side effect conundrum due to the stability of the peptides is an important concomitant secondary response to consider. At the same time, we explore the exciting new opportunities these peptides open up to us in the field of cancer therapy. The magic bullet in cancer therapy is to find a molecule that selectively affects cancer cells and not normal healthy cells. Our future studies will focus on the applications of these D peptides in *in vivo* systems under different experimental conditions. For example, future experiments will include dose-dependent peptide concentration studies on “normal” versus cancer cells to determine concentrations at which we get an effect selectively on cancer cells and not the normal cells. We have briefly discussed this in the revised manuscript (see main text page 4 and 23).